artificial intelligence/image processing/geology

carbonate, FA, porosity

**Author for correspondence:**
Honghai Kuang
e-mail: hhkuang@swu.edu.cn

# Porosity of the porous carbonate rocks in the Jingfengqiao–Baidiao area based on finite automata

## Honghai Kuang, Xi Ye and Zhiyi Qing

School of Geographic Science, Southwest University, Beibei, Chongqing, People's Republic of China

HK, 0000-0001-7997-1296

This study is based on the processing of computed microtomography images of rock samples. In this study, a finite automation is constructed using the grey value, red-green-blue (RGB) value and Euler number of polarized images of carbonate rocks from the Jingfengqiao–Baidiao area. The finite automaton is used to perform black and white binary processing of the polarized images of the carbonate rocks. The porosity of the carbonate rock is calculated based on the black and white binarization processing results of the polarized images of the carbonate rocks. The obtained porosity is compared with the carbonate porosity obtained by use of the traditional carbonate research method. When the two porosities are close, the image processing threshold of the finite automata is considered to be credible. Based on the finite automata established using the image processing threshold, the black and white binary images of the polarized images of the carbonate rocks are used to establish a rock pore image using IMAGEJ2X. The polarized images of the carbonate rocks are classified according to their RGB values using the finite automata for the porosity classification, and the obtained images are used as textures to paste onto a cube to construct a three-dimensional data model of the carbonate rocks. This study also uses 16S rDNA analysis to verify the formation mechanism of the carbonate pores in the Jingfengqiao–Baidiao area. The results of the 16S rDNA analysis show that the pores in the carbonate rocks in the Jingfengqiao–Baidiao area are closely related to microorganisms, represented by denitrifying bacteria.

## 1. Introduction

Studying the porosity of carbonate rocks is important to karst research. Therefore, it is necessary to use computer image

analysis to study the porosity of carbonate rocks. In this study, the traditional carbonate research method (TCRM) was used to verify the accuracy of the carbonate porosity results obtained using computer image analysis. The expansion of carbonate pores is sometimes related to the microorganisms in the karst water. To determine whether or not microorganisms affected the karst process in the karst water system, researchers only need to perform a 16S rDNA test.

Zhang et al. [1] discussed the image analysis method for analysing pore structures and its application to rock images. They illustrated that the black and white binarization algorithm based on the grey value is very important in the computer image analysis of rock pores. Sun et al. [2] developed a new approach to the characterization of the pore structure of shale. Their results helped to improve the algorithm used in this study. Kong et al. [3] developed a lithology recognition method based on multiresolution graph-based clustering and the K-nearest neighbour methods. Their results are very important to the lithology identification method used in this study. Wang et al. [4] conducted structural characteristic analysis of dual pore digital carbonate rocks. They showed that the image characteristics of rock pores can be used as a basis for the study of carbonate rock pores. Hua et al. [5] developed a high-resolution rock structure image processing method and applied it to carbonate reservoir evaluation. Their results helped to improve the preprocessing of polarized images of carbonate rocks. Qin et al. [6] investigated the significance of back-scattered electron images in the micro-area analysis of carbonate rocks. They showed that imaging and micro-area analysis of carbonate rocks is an effective method of studying carbonate rocks. Wang et al. [7] attempted to determine the damage fractal dimensions of a rock using scanning electron microscopy (SEM) images and MATLAB. They showed that MATLAB can be used to study carbonate pores. Ye et al. [8] studied mineral feature extraction and analysis based on multiresolution segmentation of petrographic images. Cheng et al. [9] studied rock image classification recognition based on probabilistic neural networks. They studied the acquisition of rock images of rock slides and the use of neural network algorithms to process the rock images to achieve rock image classification recognition based on probabilistic neural networks. Wang et al. [10] conducted image enhancement of rock fractures based on the fractional differential. They showed that the grey-scale algorithm is very important in the analysis of carbonate pore images. Wu et al. [11] investigated the effect of a high energy storage anchor on the surrounding rock conditions. Their results revealed that the Euler number calculation has a good prospect in the study of rock void development. Zhang et al. [12] studied the fragment extraction and repair of two-dimensional rock images based on a hybrid algorithm based on the ant colony and the Canny edge detection operator. Their results show that the hybrid algorithm based on the ant colony and the Canny edge detection operator is very important in rock image analysis. Cheng et al. [13] studied super-resolution reconstruction of rock slice images based on SINGAN. Their results show that super-resolution reconstruction of rock slice images based on SINGAN is very important in rock image analysis.

Zhang et al. [14] conducted research on the diagenesis and pore evolution of the carbonate rocks in the Nanpu area. They found that the effect of microorganisms on rock pores cannot be ignored. Lian et al. [15] proposed a new workflow for pore-type classification of carbonate reservoirs based on computed tomography (CT) images. This method is helpful in screening the carbonate pore algorithm. Lu et al. [16] conducted a visual experimental investigation of two-phase gas–water micro seepage mechanisms in fracture-cavity carbonate reservoirs. Their results are of great importance to the TCRM used in this study. Chen et al. [17] studied the pore structure characteristics and reservoir classification of the dolomite reservoirs in the fourth member of the Leikoupo Formation at the foot of the Longmen Mountains in the western Sichuan Basin. Their study sets a precedent for the traditional research methods of studying carbonate porosity. Xie et al. [18] studied the multifractality of the three-dimensional pore structures of carbonate rocks based on CT images. Their study provides a reference for the use of the Euler number to study the pores in carbonate rocks. Shou et al. [19] conducted experimental simulations of the effect of the dissolution of carbonate rocks under deep burial conditions. Their results are helpful in the experimental design of the traditional methods used to study the porosity of carbonate rocks. Wu et al. [20] explored the best threshold for binary CT imaging of carbonate rocks based on fractal theory. Wang et al. [21] conducted experimental research on the cracking of coal under temperature variations using industrial micro-CT. Dang et al. [22] investigated the application of techniques of geological remote sensing information extraction from China-Brazil Earth Resource Satellite-1 CCD data during mineral exploration. They demonstrated the importance of applying mathematical methods to the remote sensing analysis of carbonate rocks. Ge et al. [23] developed a description for rock joint roughness based on terrestrial laser scanner and image analysis. Their results show that the image acquisition equipment plays a very important role in carbonate image analysis. Liu et al. [24] studied the analysis of the strength property and pore

characteristics of the Taihang limestone using X-ray CT at high temperatures. Their results show that image analysis technology plays a very important role in carbonate pore analysis. Zhou *et al.* [25] studied the impact of water-rock interactions on the pore structures of red-bed soft rocks. Their results show that three-dimensional modelling analysis techniques play a very important role in rock pore analysis research.

Pak *et al.* [26] studied the three-dimensional imaging of a previously unidentified pore-scale process during multiphase flow in porous media. They used mathematical methods to construct a three-dimensional model of rock pores, which is a good reference for this study, in which finite automata and Euler numbers are used to construct a three-dimensional pore model of carbonate rocks. Ghiasi-Freez *et al.* [27] studied the automated Dunham classification of carbonate rocks using image processing. They used the image binary method to study the pores in rocks, which inspired the use of finite automata and Euler numbers to construct the boundary threshold of rock image pixels in this study. Goral *et al.* [28] studied the nanofabrication of synthetic nanoporous geomaterials. They used the Euler number to study rock pores, demonstrating that the Euler number can be used for carbonate rock pore research. Golreihan *et al.* [29] conducted improving preservation state assessment of carbonate microfossils in palaeontological research using label-free stimulated Raman imaging. They used image processing technology to study the pores in rocks, showing that there is a precedent for using finite automata to study the pores in rocks. Ali *et al.* [30] conducted geophysical imaging of an ophiolite structure in the United Arab Emirates.They used geoscience images to study rock pores, which inspired the use of microscopic images of rocks to study rock pores in this study. Lanari *et al.* [31] assessed XMAPTOOLS, which is a MATLAB©-based graphic user interface for microprobe quantified image processing. They used MATLAB to study rock pores, which inspired to the use of MATLAB to simulate three-dimensional rock pores in this study. Kurz [32] studied hyperspectral image analysis of different carbonate lithologies. He used the image grey value for rock image processing research, which inspired the use of rock grey image for rock pore analysis in this study. Schepp *et al.* [33] studied digital rock physics and laboratory considerations for a high-porosity volcanic rock. The results of their study provide a new idea for improving the TCRMs used in carbonate porosity research. Osorno *et al.* [34] conducted finite difference permeability calculations in large domains in a wide porosity range. Their results provide a new mathematical model for studying carbonate rock pore permeability and inspired the use of the Euler number for rock pore classification in this study. Prakash *et al.* [35] studied multi-graphics processing unit (GPU) parallelization of the maximum-likelihood expectation maximization method for digital rock tomography. Their results revealed that it is necessary to pay attention to the role of GPUs in the image analysis programming of carbonate rocks. Selem *et al.* [36] studied pore-scale imaging and analysis of low salinity water flooding in a heterogeneous carbonate rock under reservoir conditions. Their results show that the image thresholding method is very important in the study of rock pores, providing a precedent for the use of the image thresholding method in the study of rock pores. Phan *et al.* [37] assessed the use of the automatic segmentation tool for three-dimensional digital rocks using deep learning. Their results show that the automatic segmentation tool can be used for three-dimensional rock pore research, providing a good reference for the automatic segmentation tool used in this study, e.g. the finite automata and Euler number.

Seyyedi *et al.* [38] studied the pore structure changes that occur during the injection of $CO_2$ into carbonate reservoirs. They showed that the pores in carbonate rocks may indeed be affected by microbial chemical reactions. These results provide a good reference for the influence of microorganisms in karst water on carbonate rocks. Kotz *et al.* [39] studied the fabrication of arbitrary three-dimensional suspended hollow microstructures in transparent fused silica glass. They showed that the Euler number has important uses in the computer simulation of three-dimensional structures. This was very helpful to this study, in which the Euler number is used to construct a three-dimensional model of the carbonate pores. Ishutov *et al.* [40] studied the use of resin-based three-dimensional printing to build geometrically accurate proxies for porous sedimentary rocks. They showed that the Euler number classification is very important in the computer three-dimensional reconstruction of carbonate pores. This is very important to this study, in which the Euler number is used to construct the visible surface texture of the three-dimensional model of carbonate pores. Goral *et al.* [41] studied the nanofabrication of synthetic nanoporous geomaterials including nano-three-dimensional printing of digital rocks based on nanoscale-resolution three-dimensional images. They showed that a three-dimensional representation of the nano-scale carbonate structure can be achieved with the help of imaging technology. This inspired the addition of textures to only the three visible surfaces of the three-dimensional model of the rock created in this study. Berg *et al.* [42] studied industrial applications of digital rock technology. Their results provide a reminder that the algorithm used in this study must finally perform image binarization in the form of thresholds.

Yarmohammadi *et al.* [43] conducted reservoir microfacies analysis by applying microscopic image processing and classification algorithms to carbonate and sandstone reservoirs. Their results provide a reminder that polarized light microscopes can be used in the study of rock pores, i.e. in this study. Fusi *et al.* [44] assessed the use of mercury porosimetry as a tool for improving the quality of micro-CT images of low porosity carbonate rocks. Their results provide a reminder that the rock image must be preprocessed before the algorithm can be applied. Harris [45] studied a case of lanthanum carbonate ingestion, which was thought to be phlebosclerotic colitis, using CT imaging and an abdominal radiograph. His results show that there is a precedent for using finite automata in rock pore research. Maheshwari [46] compared carbonate HCl acidizing experiments with three-dimensional simulations. He showed that traditional carbonate research methods for studying carbonate rocks can be combined with computer technology. Munoz *et al.* [47] studied pre-peak and post-peak rock strain characteristics during uniaxial compression using three-dimensional digital image correlation. Their results provide a reminder that the use of image analysis technology in rock pore research can be verified using rock indicators such as the uniaxial compressive strength. Sharafisafa [48] conducted an experimental investigation of the dynamic fracture patterns of three-dimensional printed rock-like materials under impact using digital image correlation. He showed that it must be compared with the TCRMs when using imaging technology for rock research. Kim [49] conducted stress estimations via deep rock core diametric deformation and joint roughness assessment using X-ray CT imaging. Her results provide a reminder that the Euler number can be used to classify the threshold of image pixels. Saenger *et al.* [50] analysed high-resolution X-ray CT images of the Bentheim Sandstone under elevated confining pressures. Their results provide a reminder that the Euler number can be used to establish a three-dimensional model of the image pixel threshold. Gerke *et al.* [51] studied universal stochastic multiscale image fusion. Their results provide a reminder that the role of image fusion cannot be ignored when using image analysis to study rock pores. Neumann *et al.* [52] studied high accuracy capillary network representation in digital rock to reveal the permeability scaling function.

The Jinping area is surrounded by the Yalong River and contains extensive carbonate formations. As a result of the construction of hydropower projects, extensive geological surveys have been conducted in the Jinping area. To ensure the accuracy of the geological surveys, extensive drilling and sampling have been conducted in the Jinping area. The Jingfengqiao and Baidiao areas are also typical carbonate distribution areas in the Jinping area. Wells have been constructed in both locations, so deep carbonate rock samples from both locations are easy to obtain. Owing to the needs of engineering construction, the rock samples collected from these two locations require the use of TCRM for karst research. The basic parameters of the carbonate rocks obtained via karst research are good verification standards. The Baidiao area is close to the Yalong River, and it is easier to collect water samples from the Yalong River. It is convenient to use 16S rDNA testing in karst water research. Therefore, the Jingfengqiao–Baidiao area is an ideal area for studying rock porosity using microscopic images of carbonate rocks.

# 2. Experimental materials and research methods

## 2.1. Experimental materials

In this study, three rock samples from depths of 69.80–71.01 m were collected from a borehole (no: Qx404) in the Jingfengqiao area. Two water samples (BD1 and BD2) were collected in the Hebian area, Baidiao; and the water samples from Baidiao were transported to Chongqing within 27 h. The rock samples from Jingfengtqiao were transported to Chongqing within 480 h. The rock samples were collected from the $T_{2z}$ formation. The thickness of the $T_{2z}$ formation in the Jingfengqiao area is about 150–700 m. Rock sample number JP12 is griotte, with a $CaCO_3$ content of about 95%. Therefore, JP12 is a very pure carbonate rock. In this study, rock sample JP12 was cut into two rock slides. The porosity of rock sample JP12 was determined to be 0.19% using the TCRM.

## 2.2. Traditional carbonate research methods of calculating rock porosity

The TCRM of calculating rock porosity is easily understood. The rock specimen is placed in water for 72 h to soak the pores of the rock with water. After soaking the rock specimen, it is removed from the water and placed in a cool place for 1–2 h to air-dry the surface moisture. The rock specimen is weighed and placed in an oven (60°C) to dry for 24 h. After drying, the rock specimen is weighed again. The two weights are subtracted and divided by the density of water to obtain the volume of the rock pores.

Because the rock samples used in this study have other uses, all of the rock samples were standard cylinders with a diameter of 50 mm and a height of 3 mm. Thus, the volume of the rock sample was easily calculated. In this study, the porosity of the rock sample was obtained by dividing the volume of the pores by the volume of the rock sample.

## 2.3. Using finite automata to construct a three-dimensional data model of carbonate rock

The Euler number can be used to classify the pixel red-green-blue (RGB) values of rock slide images. In this study, the Euler number value of the rock image was obtained using the RGB and the following formula, and the Euler number was used to classify the colour RGB value of the rock's image pixels:

$$[E_n] = \begin{cases} \text{sech } R = \sum\limits_{n=0}^{\infty} \frac{(-1)^n}{n!(n+G+B+E_n)} \left(\frac{n}{2}\right)^{2n+G+B}, & E_n(R_n) \geq E_n(G_n), \\ \text{sech } G = \sum\limits_{n=0}^{\infty} \frac{(-1)^n}{n!(n+R+B+E_n)} \left(\frac{n}{2}\right)^{2n+R+B}, & E_n(G_n) \geq E_n(R_n), \\ \text{sech } B = \sum\limits_{n=0}^{\infty} \frac{(-1)^n}{n!(n+G+R+E_n)} \left(\frac{n}{2}\right)^{2n+G+R}, & E_n(B_n) \geq E_n(R_n), \\ \text{Grey} = R \times 0.299 + G \times 0.587 + B \times 0.114 \end{cases} \quad (2.1)$$

where $R$, $G$ and $B$ are the RGB value of the pixel; $n$ is the sequence number of the pixel; $E_n$ is the Euler number of the pixel calculated using the RGB value and the sequence number of the pixel; and Grey is the pixel's grey value calculated using the pixel's RGB value.

According to the Euler number results, the rock image was divided into nine levels according to the RGB values. In the same image of the rock, the lower the Euler number classification, the lower the porosity of the area, and the higher the Euler number classification, the higher the porosity of the area. In construction in the Jinping area, the water permeability of the rock in front of the construction is very important for construction safety. The traditional solution is to create a small artificial cave in front of the construction route. Because not all of the specimens obtained during the construction can be studied using the traditional carbonate research methods, sometimes engineers must immediately determine the porosity of the rock along the engineering route. The rock specimens collected in the small artificial cave have to be judged by engineers using the naked eye and based on experience. The accuracy of this method is not high, and it is difficult to improve the accuracy even if more engineers are recruited. Thus, as is shown in figure 1, in this study, the classified image is used as the texture of the three-dimensional cube, the classified image is obtained by classifying the RGB values of the rock image according to the Euler number, and all of the visible surfaces of the cube are displayed using the classified image to obtain a three-dimensional model of the carbonate rock. The goal of this study is that such a cube can help engineers to better judge the rock porosity.

The solution to equation (2.1) has a set of distributions [50,50]. The Euler numbers of the polarized image of the carbonate rock [50,50] were arranged in ascending order, descending order, and at random to set up a $3 \times 8$ matrix. Each row of the matrix was recorded as the spatial coordinate value to construct the nodes of the three-dimensional pore distribution model of the carbonate rock. To control the amount of calculations required, the number of nodes was kept below eight. The nodes in these cubes represent the distribution positions of the pores in the rock.

## 2.4. Finite automata image analysis method

In the TCRM, calculating the rock porosity is a three-dimensional problem. However, when the porosity of a rock slide is calculated using the image threshold analysis method, it is treated as a two-dimensional planar problem. Because the thickness of the rock slide is very thin, it can be approximated that the heights of all of the pores in the glass slide are the same (approaching 0 but not 0). Therefore, it can be approximated that the ratio of the total number of pixels representing the pores in the slide to the total number of pixels is the porosity of the slide. If an algorithm can be developed to correctly identify the rock pore pixels, the porosity can be obtained through image processing. In this study, the carbonate rocks collected from the Jingfengqiao area were processed into slides and computer images of the slides were obtained using a petrographic microscope. The images obtained using the petrographic microscope needed to be preprocessed. PHOTOSHOP was used to open the polarized image and to preprocess it into a JPG image. The JPG images followed the standard grey-scale formula and realize the grey-scale processing of the image through c# programming. All of the programming

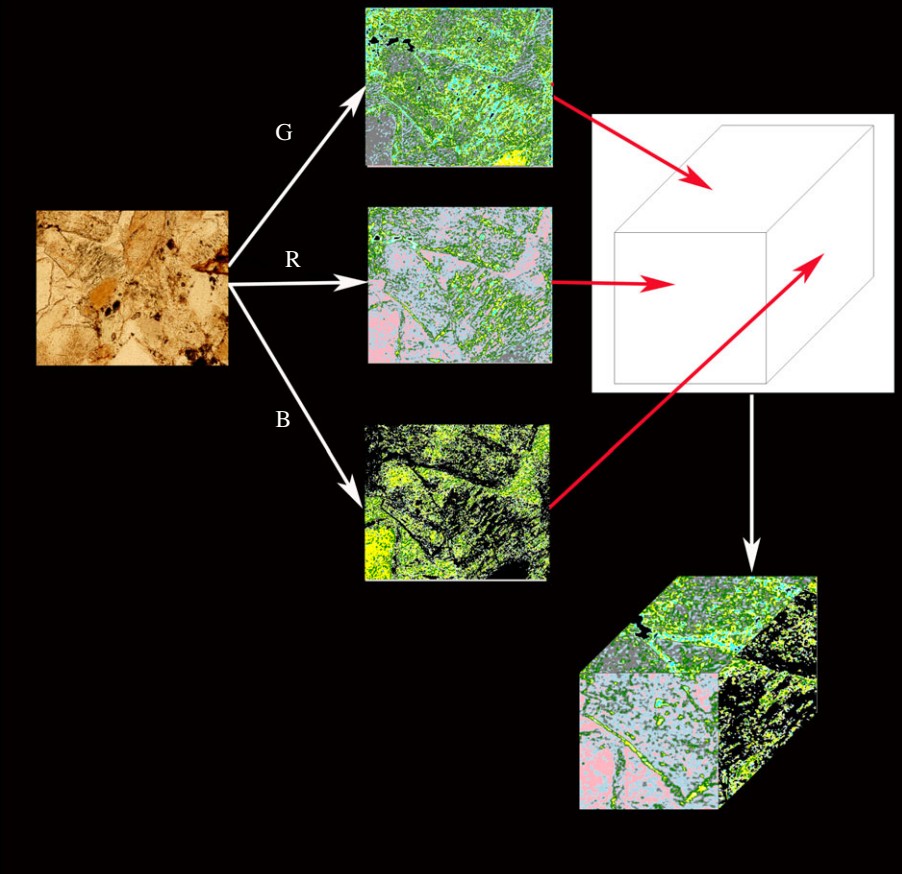

**Figure 1.** Using the image processing methods to obtain three-dimensional models of the carbonate rocks.

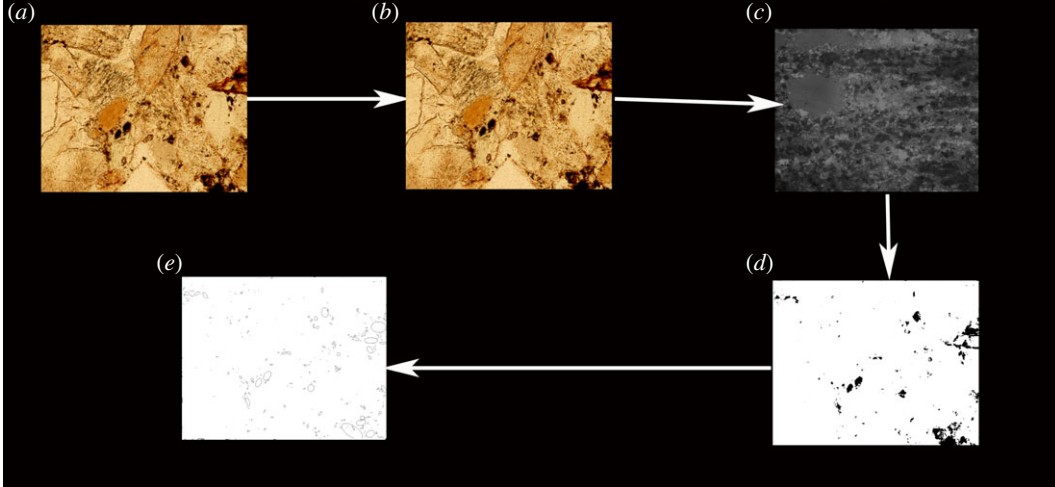

**Figure 2.** Using image processing methods to obtain the porosity of carbonate rocks. (*a*) Original image; (*b*) photoshop preprocessing of the original image; (*c*) picture after grey scale processing; (*d*) picture after finite automata image processing; and (*e*) picture after processed with ImageJ2X.

codes used in this paper have been uploaded. The image processing threshold of the polarized image was obtained by comparing it with the TCRM results. The polarized image was converted into a black and white binarized image using this image processing threshold. The porosity of the rock was obtained by dividing the number of black pixels in the polarized image after the black and white binarization processing by the total number of pixels. The porosity value was compared with the porosity value obtained using the traditional method. If the difference between the two porosities was not significant, the image processing threshold was considered to be credible. If there was a large difference between the two porosity values, the image processing threshold was not credible and

needed to be improved. The above operation was repeated until the two porosity values were close. After completing the black and white binarization, a rock pore map was obtained using IMAGEJ2X. The above process is shown in figure 2.

The characteristic of the finite automata is the finiteness of mapping. This characteristic has obvious advantages in image threshold processing. The distribution range of the image pixel RGB values is [0,255], i.e. a total of 256 mappings. In this study, Euler number was used to filter the RGB values in the finite automata, and the number of RGB values conforming to the Euler filter mapping was greatly reduced. In this study, all of the RGB values that meet the Euler number filter mapping were used as thresholds for the image threshold processing. The porosity obtained using each threshold was compared with the porosity obtained using the TCRM to determine the accuracy of the threshold. The image processing threshold mapping covered the grey value and the RGB value of the polarized images of the carbonate rocks. The grey value and RGB value corresponding to the Euler number were also used as a mapping to set the finite automata. In this study, the following finite automata was established

$$
[DFA\ M]
$$

$$
= \begin{cases}
k \in [\text{Grey}_{(i,j)}, R_{(i,j)}, G_{(i,j)}, B_{(i,j)}, E_{n_{(i,j)}}], & i \in [0,\ \text{pic1.width}],\ j \in [0,\ \text{pic1.height}] \\
\Sigma \in [T_{1(i,j)}, T_{2(i,j)}, T_{3(i,j)}, T_{4(i,j)}, T_{5(i,j)}], & f \in [q_{1(i,j)}, q_{2(i,j)}] = (0,0,0) \in \Sigma \\
T_{1(i,j)} = \text{Grey}_{(i,j)} > s,\ s \in [0,255] \in k, & T_{1(i,j)} \to A_{(i,j)},\ A_{(i,j)} = (255,255,255), \\
T_{2(i,j)} = R_{(i,j)} > s,\ s \in [0,255] \in k, & T_{2(i,j)} \to A_{(i,j)},\ A_{(i,j)} = (255,255,255) \\
T_{3(i,j)} = G_{(i,j)} > s,\ s \in [0,255] \in k, & T_{3(i,j)} \to A_{(i,j)},\ A_{(i,j)} = (255,255,255) \\
T_{4(i,j)} = B_{(i,j)} > s,\ s \in [0,255] \in k, & T_{4(i,j)} \to A_{(i,j)},\ A_{(i,j)} = (255,255,255) \\
T_{5(i,j)} = E_{n_{(i,j)}} > s,\ s \in [0,532] \in k, & E_{n_{(i,j)}} \to A_{(i,j)},\ A_{(i,j)} = (255,255,255) \\
z \in [k,f] \to \Sigma \cup [q_{1(i,j)}, q_{2(i,j)}], & z \to [(0,0,0),\ (255,255,255)] \\
q_{1(i,j)} = z + 6 + \left(1 - \frac{k}{255}\right) \times 0.5, & f = 0 \to f = 1,\ A_{(i,j)} = (255,255,255)
\end{cases}
\tag{2.2}
$$

$k$ is a finite state set; $\text{Grey}_{(i,j)}$, $R_{(i,j)}$, $G_{(i,j)}$, $B_{(i,j)}$ and $E_{n_{(i,j)}}$ are the finite state of $k$; $\Sigma$ is the map list; $T_{1(i,j)}$, $T_{2(i,j)}$, $T_{3(i,j)}$, $T_{4(i,j)}$, and $T_{5(i,j)}$ are the maps of $\Sigma$; $i$ and $j$ are the row and column number of the pixel, respectively; $f$ is the result set of $\Sigma$; $q_{(i,j)}$ is the mapping result of $\Sigma$; $A_{(i,j)}$ is the new colour value; and $z$ is the result state set.

In this study, each pixel of the carbonate polarized image was converted into a grey value or RGB value. The Euler number formula used in this study has 20 solutions to the grey value and RGB value of the carbonate polarized image. When these 20 solutions were put into [0,255], 20 image processing thresholds were obtained for the grey value or RGB value. The 20 image processing thresholds were all processed using the finite automata. For each polarized image of the carbonate rock, 20 black and white binary images were obtained using the image processing thresholds. The porosity was obtained by dividing the number of black pixels by the total number of pixels in the black and white binary image. The porosity of the rock in the polarized image must be greater than 0% and less than 100%. If the porosity is less than 0% or greater than 100%, there is an error in the finite automaton.

## 2.5. Analysis of the causes of rock pores

The results of the finite automaton image analysis method and the TCRM both show that the carbonate rock samples from the Jingfengqiao–Baidiao area have high porosities. The reason for the high porosities of the carbonate samples in the Jingfengqiao–Baidiao area is worthy of attention. The Jingfengqiao–Baidiao area is located in the Yalong River Basin. The Yalong River is the largest source of karst water in the Jingfengqiao–Baidiao area. If the Yalong River water contains microorganisms such as nitrifying bacteria, sulfobacteria and *Thiobacillus denitrificans*, the microorganisms in the Yalong River water may enter the karst water through the pores of the porous carbonate rock formations. The microorganisms in the Yalong River may produce nitrification or sulfidation in order to maintain their own survival, thereby changing the amount of hydrogen ions in the karst water, affecting the karst processes in the carbonate rock formations, and expanding the pores. To determine whether the above hypothesis is correct, water was collected from the Yalong River in the Baidiao area for 16S rDNA analysis. If microorganisms such as nitrifying bacteria, sulfobacteria and *Thiobacillus denitrificans* are detected in the Yalong River in the Baidiao area, the above hypothesis is convincing. Microbial community genomic DNA was extracted from the water samples from the Baidiao area (bd1 and bd2) using TransGen AP221-02 according to the manufacturer's instructions. The DNA extracted was checked using 1% agarose gel, and the DNA's concentration and purity were checked using NanoDrop 2000. The hypervariable region of the bacterial 16S rRNA gene was amplified using an ABIGeneAmp®9700

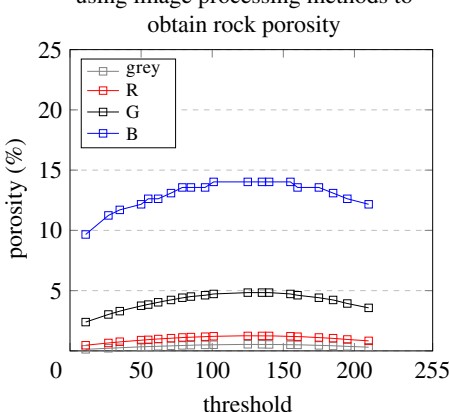

**Figure 3.** Image processing threshold curve comparison.

PCR thermocycler. The purified amplicons were pooled in equimolar and paired-end sequenced using the Illumina MiSeq PE300 platform/NovaSeqPE250 platform (Illumina, San Diego, USA) according to the standard protocols provided by Majorbio Bio-Pharm Technology Co. Ltd. (Shanghai, China).

# 3. Results

## 3.1. Image processing threshold based on finite automata

To use finite automata to study the pores in carbonate rocks, the image processing threshold of the rock image must be obtained first. In this study, different grey values and RGB values were used as the image processing thresholds to perform the black-and-white binarization of the rock images of $T_{2z}$ and to calculate the porosity. The porosity results obtained using the finite automata method were compared with the results of the traditional carbonate research method to obtain the image processing threshold. Only 21 RGB values and grey values were obtained using the finite automat. The meaningless 0 values were removed (0 cannot be used as the threshold value, if 0 is used as the threshold, it is easy to find that the RGB values or grey values of all of the pixels meet the condition or do not meet the conditions), and 20 RGB values and grey values were obtained using the finite automata. Using the 21 mapping values of the grey and RGB values corresponding to the Euler numbers as the image processing threshold, the curves shown in figure 3 were obtained.

When all four curves were placed in the same coordinate system, we found that when the value on the horizontal axis is close to the middle value of the [0,255] range, the value on the vertical axis is relatively large; and when the value on the horizontal axis is close to the two ends of the [0,255] range, the value on the vertical axis is relatively small. It was also found that the trends of the four porosity curves are exactly the same. Among them, the blue value porosity curve has the largest distribution range, while the porosity distribution range of the grey value is the smallest. The trends of the porosity curves are close because the four curves were all obtained using the same finite automaton, but the image processing thresholds used were different. By comparing the porosity obtained using the TCRM with the porosity curve, it was found that the grey value porosity curve is the closest to the results of the TCRM when the grey value is 27. According to the above comparison of the results, the grey value porosity curve is closer to the results of the TCRM than the RGB value porosity curves when the grey value is 27.

## 3.2. Results of the finite automata image analysis method

According to the above method, it was found that when the grey value was 27, the porosity was the closest to the porosity obtained using the traditional method. The black and white binary images obtained using each image processing threshold were listed, and the images shown in figures 4–7 were obtained.

As can be seen from figures 4 to 7, the processing results of the grey threshold (figure 4) shows that the difference between the different grey thresholds is relatively small, unlike the processing result of the RGB thresholds (figures 5–7), for which the difference between the different thresholds is relatively large. Figures 3–7 all show that the porosity obtained using the grey threshold is closer to the actual value measured using the TCRM than that obtained using the RGB threshold.

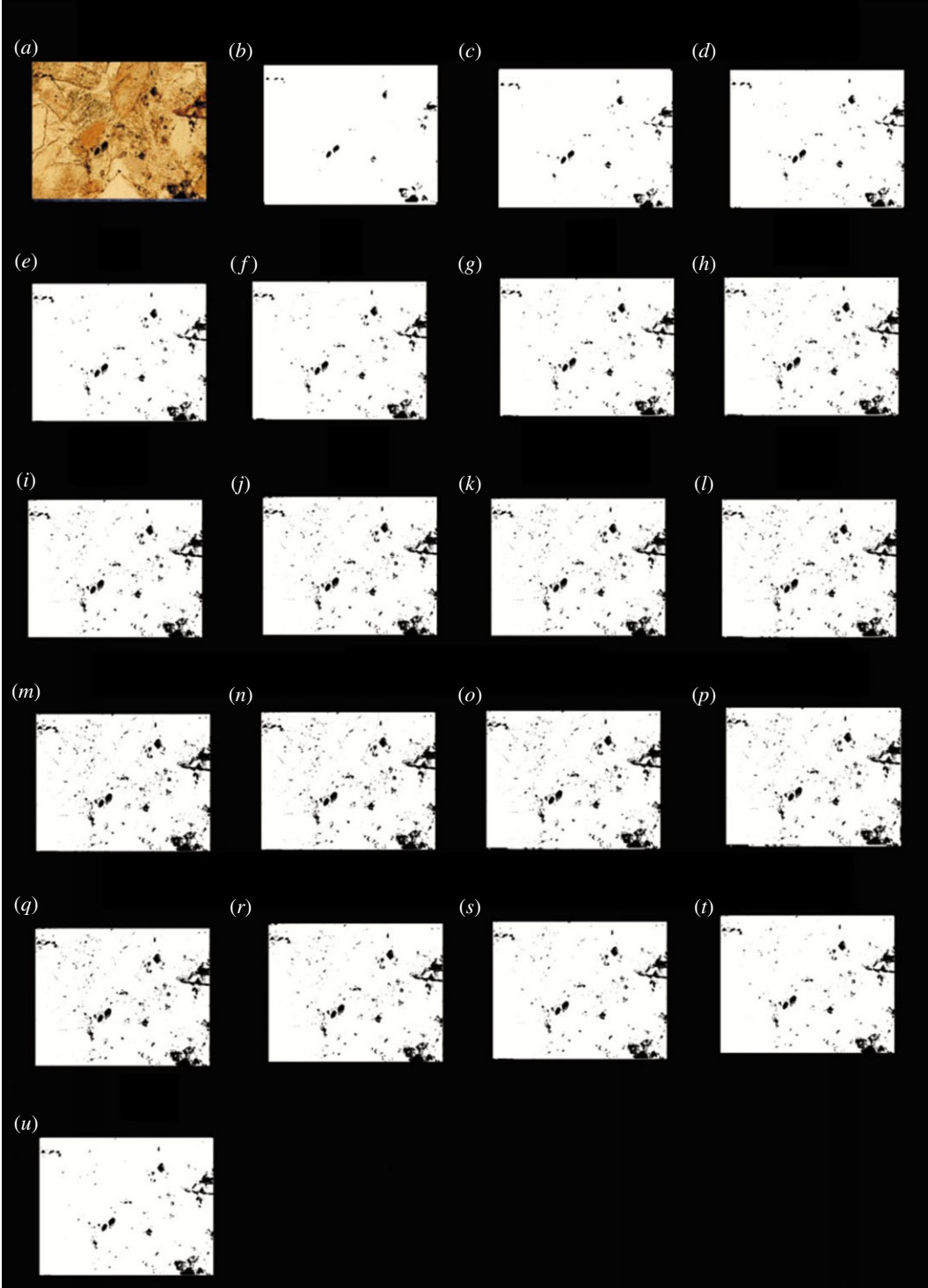

**Figure 4.** The results of the finite automata using the grey value of the images of the carbonate rocks from Jingfengqia. (*a*) Original image; and (*b*) the results of the finite automata obtained using grey value image processing thresholds of 11; (*c*) 27; (*d*) 35; (*e*) 50; (*f*) 55; (*g*) 62; (*h*) 71; (*i*) 79; (*j*) 85; (*k*) 95; (*l*) 101; (*m*) 125; (*n*) 135; (*o*) 140; (*p*) 155; (*q*) 160; (*r*) 175; (*s*) 185; (*t*) 195; and (*u*) 210.

## 3.3. Results of the 16S rDNA analysis

In this study, operational taxonomic unit (OTU) analysis was conducted to determine the microbial diversity and the abundances of the different microorganisms in the water samples from the Baidiao area. Figure 8 presents the community heatmap analysis on the phylum level. Each column represents one sample. Each row represents a phylum. The coloured blocks represent the species abundance

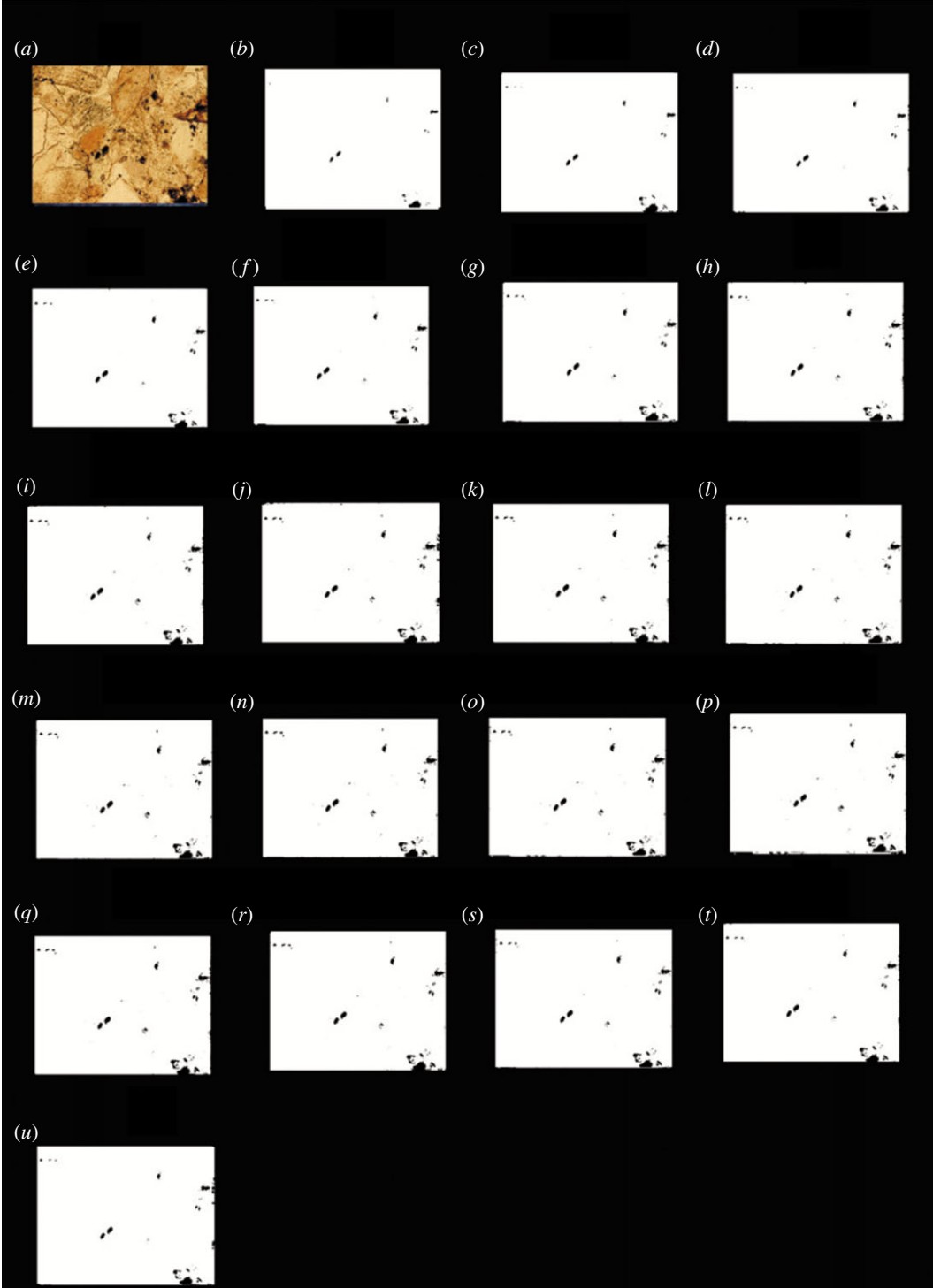

**Figure 5.** The results of the finite automata using the *R* value of the images of the carbonate rocks from Jingfengqia. (*a*) Original image; and (*b*) the results of the finite automata obtained using *R* value image processing thresholds of 11; (*c*) 27; (*d*) 35; (*e*) 50; (*f*) 55; (*g*) 62; (*h*) 71; (*i*) 79; (*j*) 85; (*k*) 95; (*l*) 101; (*m*) 125; (*n*) 135; (*o*) 140; (*p*) 155; (*q*) 160; (*r*) 175; (*s*) 185; (*t*) 195; and (*u*) 210.

values. As the legend shows, the deeper the shade of red is, the higher the abundance value is. The deeper the shade of blue is, the lower the abundance is. As can be seen from figure 8, the abundance of the unclassified-k-norank-d-Bacteria (A21) is higher in both samples. Therefore, denitrifying bacteria were present in the water samples in the Baidiao area. Based on the results obtained using the above research methods, the water samples collected from the Baidiao area do not contain nitrifying bacteria.

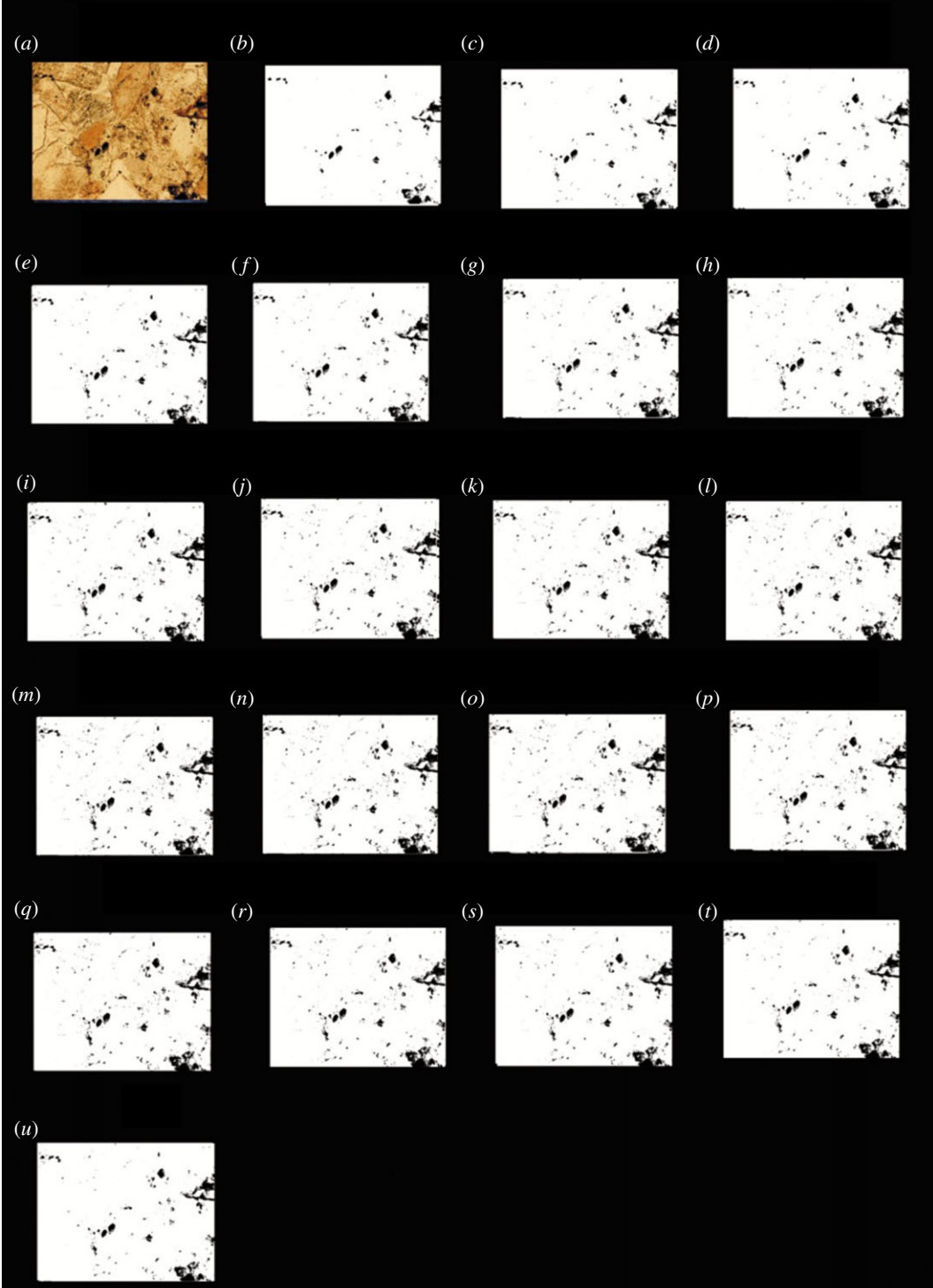

**Figure 6.** The results of finite automata using the *G* value of the images of the carbonate rocks from Jingfengqiao. (*a*) Original image; and (*b*) the results of the finite automata obtained using *G* value image processing thresholds of 11; (*c*) 27; (*d*) 35; (*e*) 50; (*f*) 55; (*g*) 62; (*h*) 71; (*i*) 79; (*j*) 85; (*k*) 95; (*l*) 101; (*m*) 125; (*n*) 135; (*o*) 140; (*p*) 155; (*q*) 160; (*r*) 175; (*s*) 185; (*t*) 195; and (*u*) 210.

## 4. Discussion and analysis

### 4.1. Comparison with the research results obtained using the traditional carbonate research method

The carbonate porosity obtained using the image threshold processing method and the carbonate porosity obtained using the TCRM were plotted on the same coordinate system and were compared

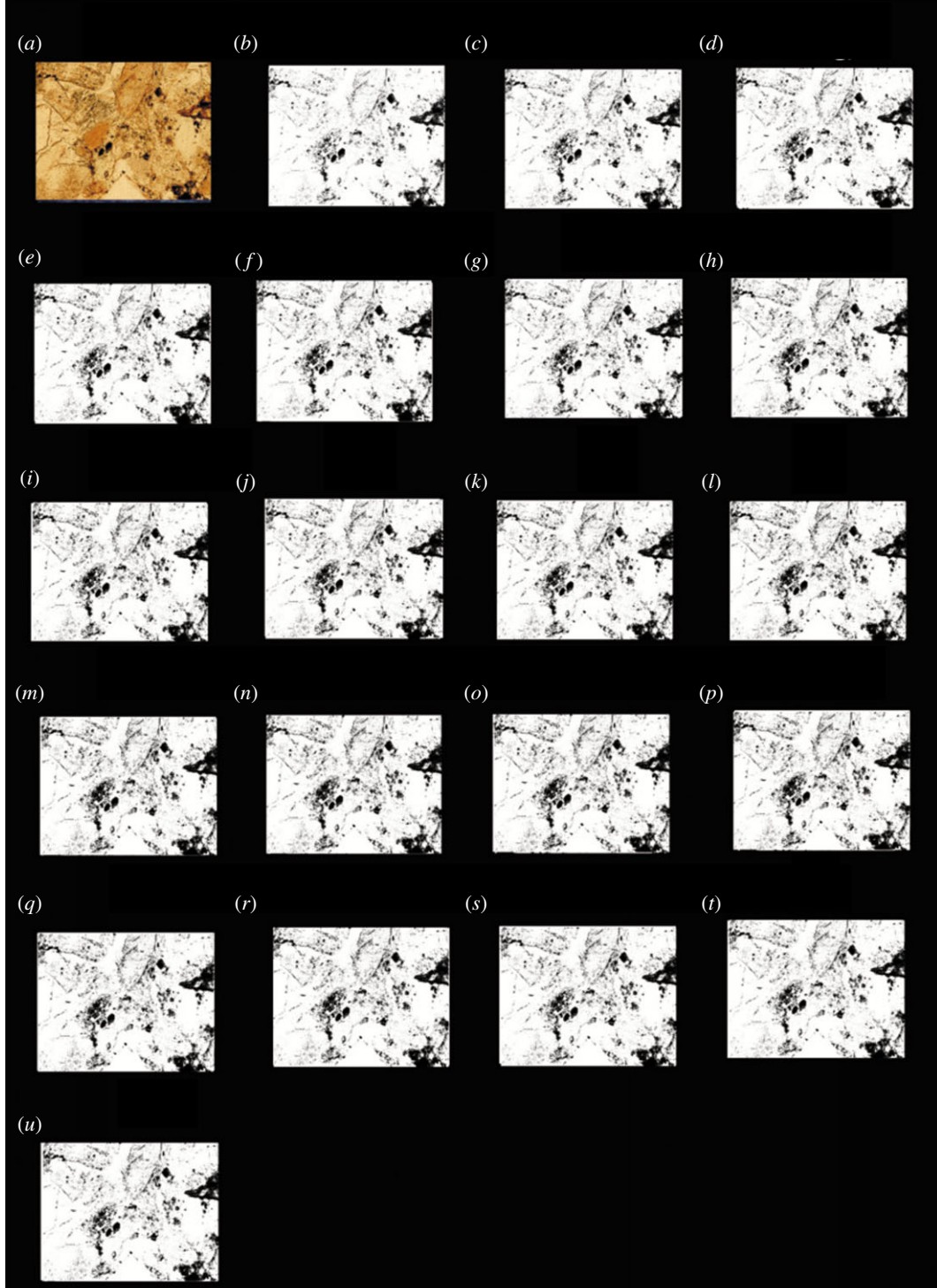

**Figure 7.** The results of the finite automata using the $B$ value of the images of the carbonate rocks from Jingfengqiao. (a) Original image; and (b) the results of the finite automata obtained using $B$ value image processing thresholds of 11; (c) 27; (d) 35; (e) 50; (f) 55; (g) 62; (h) 71; (i) 79; (j) 85; (k) 95; (l) 101; (m) 125; (n) B 135; (o) 140; (p) 155; (q) 160; (r) 175; (s) 185; (t) 195; and (u) 210.

using a histogram (figures 9 and 10). As can be seen from figure 10, the carbonate porosities of $T_{2z}$ obtained using the two methods are relatively close. The number of polarized rock images used in this study was small. If the number of polarized rock images used in the study were increased, the image processing algorithm could be improved, and the accuracy would also improve.

Numerous TCRM porosity studies have been conducted on the $T_{2z}$ formation in the Jingfengqiao. The average of the TCRM porosities obtained in previous studies is closer to the results obtained in this study.

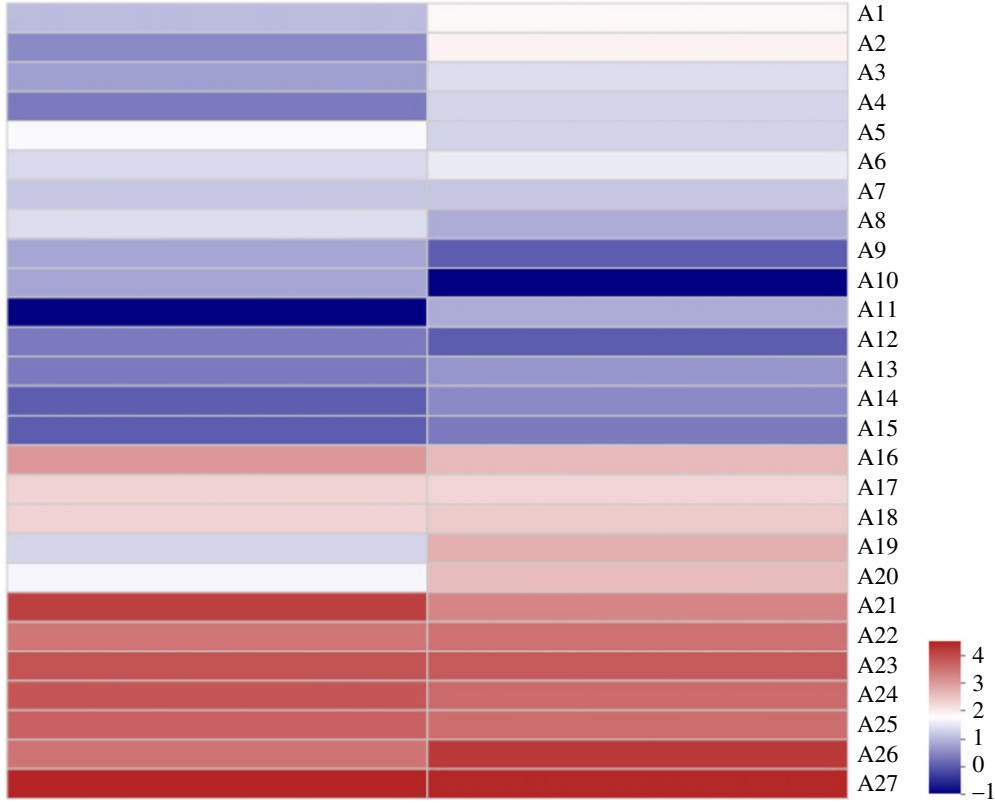

**Figure 8.** Community heat map analysis on the phylum level. A1: Gemmatimonadetes; A2: Acidobacteria; A3: Euryarchaeota; A4: Kiritimatiellaeota; A5: Nanoarchaeaeota; A6: Fibrobacteres; A7: Dependentiae; A8: Omnitrophicaeota; A9: Epsilonbacteraeota; A10: Lentisphaerae; A11: Armatimonadetes; A12: Fusobacteria; A13: Hydrogenedentes; A14: Elusimicrobia; A15: Deinococcus-Thermus; A16: Patescibacteria; A17: Chlamydiae; A18: Firmicutes; A19: Chloroflexi; A20: Planctomycetes; A21: unclassified-k-norank-d-Bacteria; A22: Verrucomicrobia; A23: Bacteroidetes; A24: Actinobacteria; A25: Spirochaetes; A26: Cyanobacteria and A27: Proteobacteria.

If the maximum and minimum TCRM porosity values of the $T_{2z}$ formation in the Jingfengqiao area are used to establish the interval, the research results of the finite automata obtained in this study are to the right of the interval.

## 4.2. Using Euler numbers to build a carbonate pore model

There are nine sets of solutions to equation (2.1), and the [50,50] set is the only set of solutions for which all of the nodes are within the cube. The distribution of the rock pores will not be outside the rock, so only the [50,50] solution was used to construct the cube in this study. As can be seen from figure 11, the cubes constructed using the Euler number nodes are greater in the $T_{2z}$ in the horizontal direction.

## 4.3. Causes of the rock pores in the carbonate rocks in the Baidiao area

The microorganisms in the Yalong River water and the microorganisms encountered when the water penetrates the surface soil enter the pores in the carbonate rocks in the Baidiao area. These microorganisms contain denitrifying bacteria. To maintain their own survival, these denitrifying bacteria will change the amount of hydrogen ions in the karst water, affecting the karst processes of the carbonate rocks in the Baidiao area. There are feldspar and pyrite in the carbonate rocks in the Baidiao area. The combination of the feldspar, pyrite, and denitrifying bacteria will change the karst processes in the carbonate formation. In summary, the nitrifying bacteria in the carbonate strata in the Baidiao area affect the karst processes in the carbonate rocks and expand the pores in the carbonate

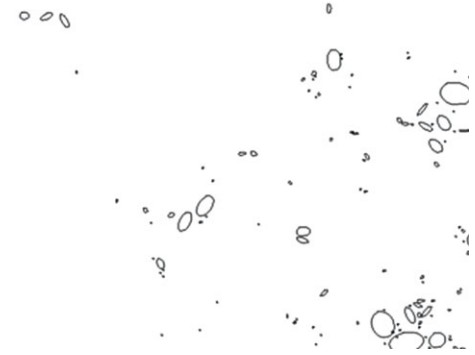

**Figure 9.** Carbonate pore map based on image threshold analysis.

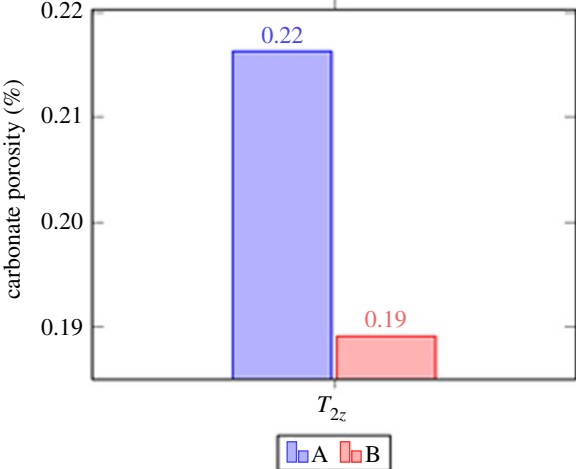

**Figure 10.** Comparison of carbonate pore value obtained by image threshold analysis method and TCRM (A: carbonate pore value obtained by image threshold analysis method; B: carbonate pore value obtained by TCRM).

rocks. Through the 16S rDNA analysis of the river water from the Baidiao area, it was found that there are denitrifying bacteria in the local Yalong River water.

## 4.4. Advantages and disadvantages of using finite automata in carbonate pore research

The advantages of using finite automata to study the pores in carbonate rocks are obvious. The processing results of the polarized images of the carbonate rocks obtained using the finite automata are very intuitive. Even people who have never learned image processing can perform pore analysis of carbonate rocks based on the results obtained from the image processing using the finite automata. Because the pixels satisfy the condition that the finite automata are finite, the quantitative calculation of the porosity can be easily performed. The calculated results can be compared with the porosity results obtained using other research methods. Compared with the research results for the Nanpu area [14], the accuracy of the results of this study is acceptable. Compared with using CT to build a three-dimensional pore model of carbonate rocks, the cost of this research method is lower. Compared with the research results for the Taihang area [24], the research method used in this study has a lower cost and a shorter research period. Compared with the study of rock pores using only image processing technology [40,41,43], the research results obtained using the TCRM to verify the image finite automata used in this study are easier for the engineering department to accept and use. The porosity results obtained using other research methods can also help improve the finite automata algorithm. The application of the finite automata to the study of carbonate pores also has obvious disadvantages. The premise of the finite automata is that the researcher must be familiar with the formal language. However, many researchers investigating carbonate rocks are not familiar with the formal language. Finite automata require correct image processing thresholds, but many researchers

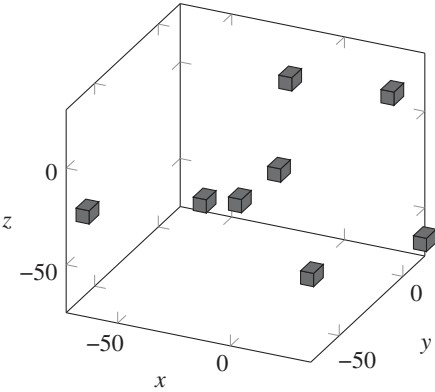

**Figure 11.** Using Euler numbers to build a carbonate pore model.

who are proficient in the formal language are not familiar with carbonate images. Thus, this requires researchers to understand both the formal language and carbonate images.

## 4.5. Can finite automata be used to study carbonate pores in other regions?

The Baidiao area in the Jinping area is surrounded by the Yalong River Basin and has a wide distribution of carbonate rocks. In the Jinping area, excluding the Baidiao area, there are many areas where studies have been conducted on the karst development, uniaxial compressive strength, and rock permeability of the carbonate rocks. These research results can be used in the study of the image processing threshold of the finite automata. Thus, the research method presented in this study can be used in the Jinping area. Owing to engineering construction, a large number of processed carbonate rock specimens for uniaxial compressive strength analysis and carbonate rock lithology analysis are available from the Jinping area. Therefore, the cost of processing the carbonate rock specimens for uniaxial compressive strength tests and carbonate rock lithology analysis is not high in the Jinping area. This is also one of the reasons why the research method presented in this study can be applied in the Jinping area. If the research method presented in this study is applied to carbonate rock areas outside of the Jinping area, it is best that the following conditions be met. The researchers have received formal language training in GEOAGENT and finite automata. Relatively long-term research has been conducted on the uniaxial compressive strength, karst development speed, and rock permeability of the local carbonate rock formation. The research data are sufficient to support the establishment of the image processing threshold. The cost of acquiring images of the local carbonate rocks is not high.

## 5. Conclusion

Finite automata was used to study polarized images of the carbonate rocks in the Jingfengqiao–Baidiao area, and the following conclusions were reached.

(i) It is feasible to use finite automata to study the porosity of the carbonate rocks in the Jingfengqiao area.

(ii) The accuracy of using the finite automata to study the pores in the polarized images of the carbonate rocks from the Jingfengqiao area is higher than that of the empirical judgement of researchers.

(iii) The method of using finite automata to study the pores in the polarized images of the carbonate rocks from the Jingfengqiao area was mainly established using the image processing threshold, which is mainly composed of the grey value, RGB value, or Euler number.

(iv) If the grey value is used to determine the image processing threshold of the finite automaton, the method of gradually approximating the results of the traditional research method is more reliable.

(v) In the Jingfengqiao area, if the Euler number is used to determine the image processing threshold of the finite automata, the final Euler number must be converted to a grey value or RGB value, which is then used as the image processing threshold of the finite automata.

(vi) The results of using the finite automata to study the pores in the polarized images of the carbonate rocks from the Jingfengqiao area show that when the grey value is used as the image processing threshold, the results are closer to the results obtained using the traditional method than the Euler number results are.

(vii) The research results for the Jingfengqiao area show that polarized images of carbonate rocks can be used to construct a three-dimensional model of the carbonate pores.

(viii) The research results for the Jingfengqiao area show that when using finite automata to study the pores in carbonate rocks, the grey values are more suitable as image processing thresholds than the RGB values.

(ix) The research results for the Jingfengqiao area show that the Euler number is a very important research tool when using finite automata to study the pores in carbonate rocks.

(x) The 16S rDNA test results of the water samples from the Baidiao area show that the local Yalong River water contains denitrifying bacteria. It is possible that the expansion of the carbonate pores in the Baidiao area originated from the microorganisms in the karst water.

(xi) The research method presented in this study can also be used in the other parts of the Jinping area, i.e. outside of the Jingfengqiao–Baidiao area.

(xii) In other regions where porous carbonate rocks are distributed, if there is a large amount of data on the rock uniaxial compressive strength, karst development speed, and rock permeability and the cost of making rock specimens and polarized glass slides is not high, the research method presented in this study can also be used in these regions.

Data accessibility. Accession to cite for these SRA data: PRJNA703089.

Authors' contributions. Figure 9 was drawn by X.Y., figure 8 was drawn by Z.Q. H.K.: conceptualization, funding acquisition, resources, software, supervision, validation, visualization, writing—original draft, writing—review and editing; X.Y.: software, writing—original draft; Z.Q.: visualization.

Competing interests. We declare we have no competing interests.

Funding. No funding has been received for the article.

Acknowledgements. We thank LetPub for its linguistic assistance during the preparation of this manuscript.

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
