## [Peer Review File · Royal Society Open Science]

Review History

RSOS-210426.R0 (Original submission)

Review form: Reviewer 1

Is the manuscript scientifically sound in its present form?

No

Are the interpretations and conclusions justified by the results?

Yes

Is the language acceptable?

Yes

Do you have any ethical concerns with this paper?

No

Have you any concerns about statistical analyses in this paper?

No

Recommendation?

Accept with minor revision (please list in comments)

Comments to the Author(s)

Dear authors,

Image processing is a promising method for porosity calculations. Thank your for your contributions. You can find my suggestions and comments below:

- Experimental Materials: Please give the depth of each sample with its formation.
- Traditional Method of calculating rock porosity: Please include oven temperature for drying process.
- Please remove Figure 3, Figures 3 and 4 have the same data.
- In Figure 4, (new Figure 3) colors of the curves should match with RGB and gray colors, please include a legend as well.
- Please do not use “we” in your text. Rephrase all those sentences in passive voice.
- Please explain Figure 12 more: what is phylum level? why are you looking at this phylum level? What is the differences between red and blue regions? Which of those bacteria are nitrifying bacteria in this list in Figure 12?
- Figures 1, 6-9, 11, 12 were not mentioned in the text. Please mention them in the manuscript.
- Under “Results of 16S rDNS analysis”: You are mentioning that” the water samples collected from the Baidiao area Do NOT contain nitrifying bacteria”, but under “(c) Causes of the rock pores in the carbonate rocks in the Baidiao area”, you are saying: “Through the 16Sr DNA analysis of the river water from the Baidiao area, it was found that there are nitrifying bacteria in the local Yalong River water”. Which one is true or is it typo mistake?
- Please edit your references based on Journal’s referencing criteria and be consistent with one criterion.
- Please use last names in citations and references, not the first names.

Thanks.

Review form: Reviewer 2

Is the manuscript scientifically sound in its present form?

Yes

Are the interpretations and conclusions justified by the results?

Yes

Is the language acceptable?

No

Do you have any ethical concerns with this paper?

No

Have you any concerns about statistical analyses in this paper?

Yes

Recommendation?

Major revision is needed (please make suggestions in comments)

Comments to the Author(s)

1. In the section of Introduction, the methods should be classified by the essence and some latest relevant references are lacking.

2. In Figure 4., you should give specific legends, otherwise, it's easy to fall into confusion.
3. In the first paragraph of page 10, the author keeps mentioning the results of traditional methods, but the paper does not find specific values or charts for comparison.
4. The formula in the paper needs to be explained to the variables.
5. In the section of Comparison with the research results obtained using traditional methods, judging from the results in Figure 13, I think there may be some errors in your description. For example, isn't 0.84 in T42y closer to 0.87? This is contrary to the description in the article.
6. In the conclusion part of this paper, the author has drawn many useful conclusions. In this paper, the carbonate porosity was obtained by image threshold processing and compared with the traditional method, the accuracy was improved to 40%. However, the final threshold still needs to be determined by approaching the traditional method, which also requires a lot of labor. Should you consider skipping this step and directly determining the threshold?
7. It is noted that your manuscript needs careful editing by someone with expertise in technical English editing paying particular attention to English grammar, spelling, and sentence structure, for helping the reader better understand the goals and results of this study.

Review form: Reviewer 3

Is the manuscript scientifically sound in its present form?

No

Are the interpretations and conclusions justified by the results?

No

Is the language acceptable?

No

Do you have any ethical concerns with this paper?

No

Have you any concerns about statistical analyses in this paper?

No

Recommendation?

Reject

Comments to the Author(s)

See the observations in the annotated PDF file of the manuscript (Appendix A).

Decision letter (RSOS-210426.R0)

Dear Dr kuang

The Editors assigned to your paper RSOS-210426 "Porosity of the Porous Carbonate Rocks in the Baidiao Area Based on Finite Automata" have made a decision based on their reading of the paper and any comments received from reviewers.

Regrettably, in view of the reports received, the manuscript has been rejected in its current form. However, a new manuscript may be submitted which takes into consideration these comments.

This decision has been taken because the referees comments and criticism are sufficiently substantial that we do not believe it will be possible for you to make the necessary responses and revisions on the time scale of a few weeks, which is what is allowed for major revision. However, given the diversity of opinion amongst the referees and the interdisciplinarity of the topic, we feel that you should have the chance to make a more thorough reworking of the paper to a form that may be more positively received by the referees.

We invite you to respond to the comments supplied below and prepare a resubmission of your manuscript. Below the referees' and Editors' comments (where applicable) we provide additional requirements. We provide guidance below to help you prepare your revision.

Please note that resubmitting your manuscript does not guarantee eventual acceptance, and we do not generally allow multiple rounds of revision and resubmission, so we urge you to make every effort to fully address all of the comments at this stage. If deemed necessary by the Editors, your manuscript will be sent back to one or more of the original reviewers for assessment. If the original reviewers are not available, we may invite new reviewers.

Please resubmit your revised manuscript and required files (see below) no later than 30-Jan-2022. Note: the ScholarOne system will 'lock' if resubmission is attempted on or after this deadline. If you do not think you will be able to meet this deadline, please contact the editorial office immediately.

Please note article processing charges apply to papers accepted for publication in Royal Society Open Science (<https://royalsocietypublishing.org/rsos/charges>). Charges will also apply to papers transferred to the journal from other Royal Society Publishing journals, as well as papers submitted as part of our collaboration with the Royal Society of Chemistry (<https://royalsocietypublishing.org/rsos/chemistry>). Fee waivers are available but must be requested when you submit your manuscript (<https://royalsocietypublishing.org/rsos/waivers>).

Thank you for submitting your manuscript to Royal Society Open Science and we look forward to receiving your resubmission. If you have any questions at all, please do not hesitate to get in touch.

on behalf of Professor Zach Agioutantis (Associate Editor) and Peter Haynes (Subject Editor)
openscience@royalsociety.org

Reviewer comments to Author:

Reviewer: 1

Comments to the Author(s)

Dear authors,

Image processing is a promising method for porosity calculations. Thank your for your contributions. You can find my suggestions and comments below:

- Experimental Materials: Please give the depth of each sample with its formation.
- Traditional Method of calculating rock porosity: Please include oven temperature for drying process.
- Please remove Figure 3, Figures 3 and 4 have the same data.
- In Figure 4, (new Figure 3) colors of the curves should match with RGB and gray colors, please include a legend as well.
- Please do not use “we” in your text. Rephrase all those sentences in passive voice.
- Please explain Figure 12 more: what is phylum level? why are you looking at this phylum level? What is the differences between red and blue regions? Which of those bacteria are nitrifying bacteria in this list in Figure 12?
- Figures 1, 6-9, 11, 12 were not mentioned in the text. Please mention them in the manuscript.
- Under “Results of 16S rDNS analysis”: You are mentioning that “ the water samples collected from the Baidiao area Do NOT contain nitrifying bacteria”, but under “(c) Causes of the rock pores in the carbonate rocks in the Baidiao area”, you are saying: “Through the 16Sr DNA analysis of the river water from the Baidiao area, it was found that there are nitrifying bacteria in the local Yalong River water”. Which one is true or is it typo mistake?
- Please edit your references based on Journal’s referencing criteria and be consistent with one criterion.
- Please use last names in citations and references, not the first names.

Thanks.

Reviewer: 2

Comments to the Author(s)

1. In the section of Introduction, the methods should be classified by the essence and some latest relevant references are lacking.
2. In Figure 4., you should give specific legends, otherwise, it's easy to fall into confusion.
3. In the first paragraph of page 10, the author keeps mentioning the results of traditional methods, but the paper does not find specific values or charts for comparison.
4. The formula in the paper needs to be explained to the variables.
5. In the section of Comparison with the research results obtained using traditional methods, judging from the results in Figure 13, I think there may be some errors in your description. For example, isn't 0.84 in T42y closer to 0.87? This is contrary to the description in the article.
6. In the conclusion part of this paper, the author has drawn many useful conclusions. In this paper, the carbonate porosity was obtained by image threshold processing and compared with the traditional method, the accuracy was improved to 40%. However, the final threshold still needs to be determined by approaching the traditional method, which also requires a lot of labor. Should you consider skipping this step and directly determining the threshold?
7. It is noted that your manuscript needs careful editing by someone with expertise in technical English editing paying particular attention to English grammar, spelling, and sentence structure, for helping the reader better understand the goals and results of this study.

Reviewer: 3

Comments to the Author(s)

See the observations in the annotated PDF file of the manuscript ("RSOS-210426_Proof_hi_rev.pdf").

===PREPARING YOUR MANUSCRIPT===

===PREPARING YOUR REVISION IN SCHOLARONE===

<https://royalsociety.org/journals/authors/author-guidelines/#supplementary-material> to include a suitable title and informative caption. An example of appropriate titling and captioning may be found at https://figshare.com/articles/Table_S2_from_Is_there_a_trade-off_between_peak_performance_and_performance_breadth_across_temperatures_for_aerobic_sc_ope_in_teleost_fishes_/3843624.

Author's Response to Decision Letter for (RSOS-210426.R0)

See Appendix B.

Decision letter (RSOS-211844.R0)

Dear Dr kuang,

I am pleased to inform you that your manuscript entitled "Porosity of the Porous Carbonate Rocks in the Jingfengqiao-Baidiao Area Based on Finite Automata" is now accepted for publication in Royal Society Open Science. [The view of Prof Agioutantis, as Associate Editor, and myself, as Subject Editor, was that you have made a significant effort to address the many comments of the reviewers and that, whilst the paper could be improved further, it is now much more suitable for publication than it was and little will be gained by a further round of reviews.]

on behalf of Professor Zach Agioutantis (Associate Editor) and Peter Haynes (Subject Editor)
openscience@royalsociety.org

Follow Royal Society Publishing on Twitter: @RSocPublishing
Follow Royal Society Publishing on Facebook:
<https://www.facebook.com/RoyalSocietyPublishing.FanPage/>

Read Royal Society Publishing's blog:
<https://royalsociety.org/blog/blogsearchpage/?category=Publishing>

Appendix A**ROYAL SOCIETY
OPEN SCIENCE****Porosity of the Porous Carbonate Rocks in the Baidiao Area
Based on Finite Automata**

Journal:	Royal Society Open Science
Manuscript ID	RSOS-210426
Article Type:	Research
Date Submitted by the Author:	17-Mar-2021
Complete List of Authors:	kuang, honghai; Southwest University, Ye, Xi ; Southwest University, school of geographic science Qing, Zhiyi ; Southwest University, school of geographic science
Subject:	Artificial intelligence < COMPUTER SCIENCE, Image processing < COMPUTER SCIENCE, Geology < EARTH SCIENCES
Keywords:	Karst, FA, carbonate
Subject Category:	Earth and Environmental Science
Note: The following files were submitted by the author for peer review, but cannot be converted to PDF. You must view these files (e.g. movies) online.	
upload_3-7-17.rar	

Author-supplied statements

Relevant information will appear here if provided.

Ethics

Does your article include research that required ethical approval or permits?:

This article does not present research with ethical considerations

Statement (if applicable):

CUST_IF_YES_ETHICS :No data available.

Data

It is a condition of publication that data, code and materials supporting your paper are made publicly available. Does your paper present new data?:

Yes

Statement (if applicable):

Accession to cite for these SRA data: PRJNA703089;

it will be accessible with the following link :

<https://www.ncbi.nlm.nih.gov/sra/PRJNA703089>

source code and images: Dryad doi:10.5061/dryad.t76hdr80n

I share my unpublished dataset using this temporary link: https://datadryad.org/stash/share/FAI-ohaWxKFJ3L--Qk5DwGM1842I8V_VgXheYDc8uOI.

Conflict of interest

I/We declare we have no competing interests

Statement (if applicable):

CUST_STATE_CONFLICT :No data available.

Authors' contributions

This paper has multiple authors and our individual contributions were as below

Statement (if applicable):

Honghai Kuang carried out the source code, participated in data analysis, carried out sequence alignments, participated in the design of the study and drafted the manuscript;

Xi Ye carried out the statistical analyses and Fig 9 was drawn by Xi Ye;

Zhiyi Qing conceived of the study, designed the study, and Fig 12 was drawn by Zhiyi Qing.

All authors gave final approval for publication.

ROYAL SOCIETY
OPEN SCIENCE

rsos.royalsocietypublishing.org

Research

Article submitted to journal

Subject Areas:

Karst, FA, rock

Keywords:

carbonate, FA, porosity

Author for correspondence:

Honghai Kuang

e-mail: hhkuang@swu.edu.cn

The text needs an extensive revision
of the English language -- (see the annotations)

Experimental methods for porosity determination?

Which classification was applied?

You should mention that the study is based on the
processing of computerized microtomography images of
rock samples.

This method is not applied for porosity measurements in
rock samples by the industry. The porosity is usually
measured through controlled liquid or gas injection in the
rocks, and density values can be used to calibrate porosity
essays.

Porosity of the Porous Carbonate Rocks in the Baidiao Area Based on Finite Automata

Honghai Kuang¹, Xi Ye¹, Zhiyi Qing¹

¹school of geographic science, southwest
university, beibei, chongqing, PR china

In this study, the gray value, RGB value, and Euler number of the polarized images of the carbonate rocks in the Baidiao area are used to construct a finite automaton and the finite automaton is used to perform black and white binary processing on the polarized images of the carbonate rocks. The porosity of the carbonate rock is calculated based on the black and white binarization processing results of the polarized images of the carbonate rocks. The obtained porosity is compared with the carbonate porosity obtained via traditional research methods. When the two porosities are close, the image processing threshold of the finite automata is considered to be credible. Based on the finite automata established using the image processing threshold, the black and white binary images of the polarized images of the carbonate rocks are used to establish a rock pore map. The polarized images of the carbonate rocks are classified according to their RGB values using the finite automata for porosity classification, and the obtained images are used as textures to be pasted onto the cube to construct a 3D data model of the carbonate rocks. This paper also uses 16S rDNA analysis to verify the formation mechanism of the carbonate pores in the Baidiao area. The results of the 16S rDNA analysis show that the pores in the carbonate rocks in the Baidiao area are closely related to microorganisms represented by nitrifying bacteria.

1. Introduction

The study of the porosity of carbonate rocks is important to karst investigations. The cost of the widely used dry and wet weighing method to study rock porosity is relatively high. Therefore, it is necessary to use computer image analysis to study the porosity of carbonate rocks.

© 2014 The Authors. Published by the Royal Society under the terms of the Creative Commons Attribution License <http://creativecommons.org/licenses/by/4.0/>, which permits unrestricted use, provided the original author and source are credited.

THE ROYAL SOCIETY
PUBLISHING

~~To study the porosity of carbonate rocks using computer image analysis, we must first choose an appropriate algorithm. If the computer image analysis algorithm is not accurate, the reliability of the results of the carbonate porosity study will be questionable. It is best to use traditional research methods to verify the accuracy of the carbonate porosity results obtained using computer image analysis.~~ The expansion of carbonate pores is sometimes related to microorganisms in karst water. ~~In order to determine whether or not there are microorganisms that affect the karst process in karst water, researchers only need to perform a 16S rDNA test.~~

Jiquan et al. discussed the image analysis method of analyzing the pore structure and its application to rock images[1]. They illustrated that the black and white binarization algorithm based on the gray value is very important in the computer image analysis of rock pores. Wenfeng et al. found a new approach to the characterization of the pore structure of shale[2]. Their results helped improve the algorithm used in this study. Jiazheng et al. conducted research on the diagenesis and pore evolution of carbonate rocks in the Nanpu area[3]. They found that the effect of microorganisms on rock pores cannot be ignored. ~~Lian et al. proposed a new workflow for the pore-type classification of carbonate reservoirs based on computed tomography (CT) images[4]. This method is helpful for screening the carbonate pore algorithm. Kong et al. developed a lithology recognition method based on multi-resolution graph-based clustering and the K-nearest neighbor[5]. Their results are very important to the lithology identification method used in this study.~~ Lu et al. conducted a visual experimental investigation of two-phase gas-water microseepage mechanisms in fracture-cavity carbonate reservoirs[6]. Their results are of great help to the traditional research methods used in this study. Yulin et al. studied the pore structure characteristics and reservoir classification of the dolomite reservoirs in the fourth member of the Leikoupo Formation, at the foot of Longmen Mountain, western Sichuan Basin[7]. Their study provides a precedent for the traditional research methods of studying carbonate porosity. Shuyun et al. studied the multifractality of the 3D pore structures of carbonate rocks based on CT images[8]. ~~This study provides a reference for the use of the Euler number to study the pores in carbonate rocks.~~ Chenchen et al. conducted structural characteristic analysis of carbonate dual pore digital rock[9]. They showed that the image characteristics of rock pores can be used as a basis for the study of carbonate rock pores. Jianfeng et al. conducted experimental simulations of the effect of the dissolution of carbonate rock under deep burial conditions[10]. Their results are helpful for the experimental designs of the traditional methods used to study the porosity of carbonate rocks. Guoming et al. explored the best threshold for binary CT imaging of carbonate rocks based on fractal theory[11]. ~~They confirmed that carbonate porosity can be obtained through image binarization operation.~~ Dengke et al. conducted experimental research on the cracking process of coal under temperature variations using industrial micro-CT[12]. ~~They demonstrated that microscopic techniques can be used for rock pore research.~~ Hua et al. developed a high-resolution rock structure image processing method and applied it to carbonate reservoir evaluation[13]. Their results helped improve the preprocessing of carbonate polarized images. Yujuan et al. investigated the significance of a back-scattered electron image in the micro-area analysis of carbonate rocks[14]. They showed that imaging and micro-area analysis of carbonate rocks is an effective method for carbonate rock research. Fenge et al. attempted to determine the damage fractal dimensions of rock scanning electron microscopy (SEM) images in the Matlab environment[15]. They showed that MATLAB can be used in the study of carbonate pores. Runqing et al. studied mineral feature extraction and analysis based on multiresolution segmentation of petrographic images[16]. ~~They showed that the use of Euler numbers must be emphasized in the study of carbonate pores.~~ Guojian et al. studied rock image classification recognition based on probabilistic neural networks[17]. They illustrated the importance of neural network algorithms in the study of carbonate slides. Weixing et al. carried out image enhancement of rock fractures based on the fractional differential[18]. They showed that the gray scale algorithm is very important in the analysis of carbonate pore images. Fuxing et al. investigated the application of techniques for geological remote sensing information extraction from CBERS-1 CCD data in mineral exploration[19]. They demonstrated

Confusing sentence

It is a simplistic assumption. The occurrence of bacteria in the pore water does not automatically mean that the dissolution was

what traditional method?

???????

what is the context?

the importance of applying mathematical methods to the remote sensing analysis of carbonate lithology.

Pak et al. studied the 3D imaging of a previously unidentified pore-scale process during multiphase flow in porous media[20]. They showed that it is feasible to use mathematical methods to simulate the pore distribution of carbonate rocks in 3D. Seyyedi et al. studied the pore structure changes that occur during the injection of CO₂ into carbonate reservoirs[21]. They showed that the pores in carbonate rocks may indeed be affected by microbial chemical reactions. Ghiasi-Freez et al. studied the automated Dunham classification of carbonate rocks using image processing[22]. They showed that it is feasible to apply binary image analysis to the study of carbonate pores. Goral et al. studied the nanofabrication of synthetic nanoporous geomaterials[23]. They showed that the Euler number plays an important role in the study of rock pores. Kotz et al. studied the fabrication of arbitrary 3D suspended hollow microstructures in transparent fused silica glass[24]. They showed that the Euler number has important uses in the computer simulation of 3D structures. Ishutov et al. studied the use of resin-based 3D printing to build geometrically accurate proxies for porous sedimentary rocks[25]. They showed that Euler number classification is very important in the computer 3D reconstruction of carbonate pores. Golreihan et al. conducted improving preservation state assessment of carbonate microfossils in paleontological research using label-free stimulated Raman imaging[26]. They showed that image processing technology has a wide range of applications in carbonate rock research. Goral et al. studied the nanofabrication of synthetic nanoporous geomaterials from nanoscale-resolution 3D imaging to nano-3D-printed digital rocks[27]. They showed that 3D demonstration of the nano-scale carbonate structure can be achieved with the help of imaging technology. Berg et al. studied industrial applications of digital rock technology[28]. They showed that digital imaging technology can be applied to the field of carbonate pore analysis. Ali et al. conducted geophysical imaging of an ophiolite structure in the United Arab Emirates[29]. They showed that geoscience images can be used to study the internal structures of rocks. Lanari et al. studied XMapTools, which is a Matlab©-based graphic user interface for microprobe quantified image processing[30]. They demonstrated the important role that MATLAB in image processing. Yarmohammadi et al. conducted reservoir microfacies analysis

exploiting microscopic image processing and classification algorithms applied to carbonate and sandstone reservoirs[31]. They showed that it is necessary to apply polarized light microscopy in the study of carbonate rocks. Kurz studied hyperspectral image analysis of different carbonate lithologies[32]. He showed that the application of image grayscale technology to carbonate rock research is feasible. Fusi et al. studied mercury porosimetry as a tool for improving the quality of micro-CT images of low porosity carbonate rocks[33]. They showed that it is feasible to use image technology from multiple sources in carbonate research. Harris studied a case of lanthanum carbonate ingestion thought to be phlebosclerotic colitis using CT imaging and abdominal radiograph[34]. He showed that the image finite automata algorithm is very important to carbonate rock research. Maheshwari compared carbonate HCl acidizing experiments with 3D simulations[35]. He showed that the traditional methods of studying carbonate rocks can be combined with computer technology. Munoz et al. studied pre-peak and post-peak rock strain characteristics during uniaxial compression using 3D digital image correlation[36]. They showed that image technology can be used to study the uniaxial compressive strength of rocks. Sharafisafa conducted an experimental investigation of the dynamic fracture patterns of 3D printed rock-like materials under impact using digital image correlation[37]. They showed that it must be compared with traditional experimental methods when using imaging technology for rock research. Kim conducted stress estimations via deep rock core diametrical deformation and joint roughness assessment using X-ray CT imaging[38]. She showed that the Euler number can be classified using image processing technology in rock pore research. Schepp et al. studied digital rock physics and laboratory considerations for a high-porosity volcanic rock[39]. Their study provides a new idea for improving the traditional method of carbonate porosity research. Saenger et al. analyzed high-resolution X-ray computed tomography images of the Bentheim sandstone under elevated confining pressures[40]. Their results provide a new image processing

algorithm model. Osorno et al. conducted finite difference calculations of permeability in large domains in a wide porosity range[41]. Their results provide a new mathematical model for the study of carbonate rock pore permeability.

The Jinping area is surrounded by the Yalong River and contains extensive carbonate formations. The Baidiao area is a typical area containing carbonate rocks in the Jinping area. The local carbonate stratum has a deep burial depth and a large number of pores. A uniaxial pressure test conducted on the local carbonate found that the porosities of the rock samples from many of the formations increased and the uniaxial pressure resistance decreased. The decrease in the uniaxial compressive strength of the local carbonate rocks may be caused by microorganisms in the karst water. Whether this conjecture is correct or not can be determined via 16S rDNA testing. The uniaxial compressive strength of the local rocks is closely related to their porosity. Therefore, it is necessary to develop a low-cost method for researching carbonate pores.

You should provide more information on the geological setting. The text is confusing and need to be completely rewritten.

The information about the uniaxial test and the influence of microorganisms on the variation of the rock mechanics is confusing. This sections contains information about the results? And conclusion?

2. Experimental materials and research methods

(a) Experimental materials

In this study, 11 rock samples and 2 water samples (BD1 and BD2) were collected in the Hebian area, Baidiao, and were transported to Chongqing within 27 hours. The 11 rock samples were collected from different formations. Three of the rock samples were broken into pieces during processing. Eight of the rock samples were successfully processed into the required shape. Five of the rock samples were successfully processed into glass slides.

Samples were collected from well cores, during drilling operations?

What are the formations from which the samples came? What is the description of these carbonate rocks?

What is the "required shape" of the samples? Glass slides refers to thin sections? How the thin sections were prepared?

(i) Traditional method of calculating rock porosity

The traditional method of calculating rock porosity is easily understood. The rock specimen is placed in water for 72 hours to soak the pores of the rock with water. After soaking the rock specimen, it is taken out of the water and placed in a cool place for 1–2 hours to air-dry the surface moisture. The rock specimen is weighed and placed in an oven to dry for 24 hours. After drying, the rock specimen is weighed again. The two weights are subtracted and divided by the density of water to obtain the volume of the rock pores. Because the rock samples used in this study have other uses, all of the rock samples were standard cylinders with a diameter of 50 mm and a height of 3 mm. Thus, the volume of the rock sample was easily calculated. In this study, the porosity of the rock sample was obtained by dividing the volume of the pores by the volume of the rock sample.

This is not a standard method used by the industry to quantify porosity in rock samples. It is used to verify density, and by the concept can be used to estimate the volume of empty space of rock samples. However this is inaccurate for porosity measurements in rocks with microporosity and complex capilarity.

(b) Using finite automata to construct a 3D data model of carbonate rock

It is no clear if the work used thin section images. uCT images...?

The Euler number can be used to classify the pixel RGB values of the polarized image of the rock. In this study we used the following formula to obtain the Euler number using the RGB value of the rock image, and the Euler number was used to classify the color RGB value of the rock's image pixels:

$$[E_n] = \begin{cases} sechR = \sum_{n=0}^{\infty} \frac{(-1)^n}{n!(n + G + B + E_n)} \left(\frac{n}{2}\right)^{2n+G+B}, & E_n(R_n) \geq E_n(G_n), \\ sechG = \sum_{n=0}^{\infty} \frac{(-1)^n}{n!(n + R + B + E_n)} \left(\frac{n}{2}\right)^{2n+R+B}, & E_n(G_n) \geq E_n(R_n), \\ sechB = \sum_{n=0}^{\infty} \frac{(-1)^n}{n!(n + G + R + E_n)} \left(\frac{n}{2}\right)^{2n+G+R}, & E_n(B_n) \geq E_n(R_n), \\ Gray = R \times 0.299 + G \times 0.587 + B \times 0.114 \end{cases} \quad (2.1)$$

According to the Euler number results, the rock polarized image was divided into nine levels according to the RGB value. In the same polarized image of the rock, the lower the Euler number classification, the lower the porosity of the area, and the higher the Euler number classification, the higher the porosity of the area. **The classified image was used as the texture of the 3D cube and was pasted it onto each surface of the cube, and then, 3D models of the carbonate rock could be obtained.**

If the samples are cylindrical, why the work tried to build a cubic model? The images were obtained from thin sections, and after that processed to reproduce a cube? What is the rationale applied here? This stage of the work/methodology is completely missing.

Figure 1. Using the image processing methods to obtain 3D models of the carbonate rocks

The graph is confusing. Please inform the scale of the images, the pixel scale, the are imaged...?

This figure is not mentioned in the text above(?)

The graph does not make clear how the treated images (R, G and B channels of each image) are used to populate the cube?

(c) Finite automata image analysis method

In this study, the carbonate rocks collected from the Baidiao area were processed into glass slides and computer images of the slides were obtained using a polarizing microscope. The image obtained using the polarizing microscope needed to be preprocessed. Photoshop was used to open the polarized image and to preprocess it into a JPG image. The JPG images were converted to grayscale images through programming. The image processing threshold of the polarized image was obtained by comparing it with the traditional research methods. The polarized image was converted into black and white binarization using this image processing threshold. The porosity of the rock was obtained by dividing the number of black pixels in the polarized image after the black and white binarization processing by the total number of pixels. The porosity value was compared with the porosity value obtained using the traditional method. If the difference between the two porosities was not significant, the image processing threshold was considered to be credible. If there was a big difference between the two porosity values, the image processing threshold was not credible and needed to be improved. The above operation was repeated until the two porosity values were close. After completing the black and white binarization, a rock pore map was obtained with the help of imagej2x. The above process is shown in Figure 2.

Please use numbering of the images to provide a description of each processing stage. What is the scale of the images (pixel scale).

Figure 2. Using image processing methods to obtain the porosity of carbonate rocks

You need to explain the procedures for the threshold definition

In order to ensure the correctness of the finite automata, multiple image processing threshold mappings were used when establishing the finite automata. The image processing threshold mapping covered the gray value and the RGB value of the polarized images of the carbonate rocks. The gray value and RGB value corresponding to the Euler number were also used as a mapping to set the finite automata. In this study, the following finite automata was established:

$$[DFA \quad M] = \left\{ \begin{array}{l} k \in [gray(i,j), R(i,j), G(i,j), B(i,j), En(i,j)], \quad i \in [0, pic1.width], j \in [0, pic1.height] \\ \sum \in [T_1(i,j), T_2(i,j), T_3(i,j), T_4(i,j), T_5(i,j)], \quad f \in [q_1(i,j), q_2(i,j)] = (0, 0, 0) \in \sum \\ T_1(i,j) = gray(i,j) > s, s \in [0, 255] \in k, \quad T_1(i,j) \rightarrow A(i,j), A(i,j) = (255, 255, 255) \\ T_2(i,j) = R(i,j) > s, s \in [0, 255] \in k, \quad T_2(i,j) \rightarrow A(i,j), A(i,j) = (255, 255, 255) \\ T_3(i,j) = G(i,j) > s, s \in [0, 255] \in k, \quad T_3(i,j) \rightarrow A(i,j), A(i,j) = (255, 255, 255) \\ T_4(i,j) = B(i,j) > s, s \in [0, 255] \in k, \quad T_4(i,j) \rightarrow A(i,j), A(i,j) = (255, 255, 255) \\ T_5(i,j) = En(i,j) > s, s \in [0, 532] \in k, \quad En(i,j) \rightarrow A(i,j), A(i,j) = (255, 255, 255) \\ z \in [k, f] \rightarrow \sum \cup [q_1(i,j), q_2(i,j)], \quad z \rightarrow [(0, 0, 0), (255, 255, 255)] \\ q_1(i,j) = z + 6 + (1 - \frac{k}{255}) \times 0.5, \quad f = 0 \rightarrow f = 1, A(i,j) = (255, 255, 255) \end{array} \right. \quad (2.2)$$

The text need to explain all the terms used in the Equations!

(d) Analysis of the causes of rock pores

The results of the finite automaton image analysis method and the traditional research methods both showed that the samples of the carbonate formations in the Baidiao area have high porosities. The reason for the high porosities of the samples of the carbonate strata in the Baidiao area is worthy of attention. The Baidiao area is located in the Yalong River Basin. The Yalong River is the largest source of karst water in the Baidiao area. **If the Yalong River water contains microorganisms such as nitrifying bacteria, sulfobacteria, and *Thiobacillus denitrificans*, the microorganisms in the Yalong River water may enter the karst water through the pores of the porous carbonate rock formations.** The microorganisms in the Yalong River may produce nitrification or sulfidation in order to maintain their own survival, thereby changing the amount of hydrogen ions in the karst water, affecting the karst process of the carbonate rock formations, and expanding the pores. **To determine whether the above hypothesis is correct, we collected water from the Yalong River in the Baidiao area for 16S rDNA analysis. If microorganisms such as nitrifying bacteria, sulfobacteria, and *Thiobacillus denitrificans* are detected in the Yalong River in the Baidiao area, the above hypothesis is convincing.**

What was the depth of sampling of the studied rocks? The assumption that superficial waters enters the studied formations need to be better addressed. The formulation of the hypothesis about the relationship between the presence of bacteria in the superficial waters and the formation of pores in the buried carbonate rocks is very confusing, and it has no relation with the other part of the research. Furthermore, finding bacteria which could be associated with carbonate dissolution in superficial waters does not means that the porosity was created only by their activity in formation waters.

3. Results

(a) Image processing threshold based on finite automata

To use finite automata to study the pores in carbonate rocks, the image processing threshold of the rock image must be obtained first. In this study, different gray values and RGB values were used as the image processing thresholds to perform the black-and-white binarization of the **rock images of T_{2y}^4 and to calculate the porosity.** The porosity results obtained using the finite automata method were compared with the results of the traditional method to obtain the image processing threshold. **By using the 21 mapping values of the gray and RGB values corresponding to the Euler numbers as the image processing threshold, the curves shown in Figure 3 were obtained.**

Explain the terms used in the equations and discretization

Figure 3. Using image processing methods to obtain the porosity of carbonate rocks

The explanation is very confusing. The text concluded that the thresholds are not accurate for the porosity determination.

Figure 3 shows the porosity curves obtained using the gray value and RGB value separately. We found that although the porosity distribution ranges of the porosity curves are different, the trends of the four porosity curves are consistent. The porosity distribution ranges of the four porosity curves are very large, indicating that the image processing threshold of the finite automata must use porosity research results obtained using other methods as a reference. At this time, the porosity research results obtained via traditional methods are a good reference. By putting the four porosity curves together, we obtained Figure 4.

Be clear - the analysis of an image obtained from a thin section can not establish the porosity of a rock. The estimation should be based on tens or hundreds of images from a large number of thin sections. It depends on the nature of the rock, its texture, nature of porosity and the thickness of the interval of interest.

Specify which color is linked to each parameter.

Figure 4. Image processing threshold curve comparison

Which trend?

We found that the trends of the four porosity curves are exactly the same. Among them, the R value porosity curve has the largest distribution range and porosity distribution range of the B value is the smallest. The distribution range of the G value porosity curve is larger than that of the gray value porosity curve. The trends of the porosity curves are close because the four curves were all obtained using the same finite automaton, but the image processing thresholds used were different. By comparing the porosity obtained using the traditional method with the porosity curve, it was found that the gray value porosity curve is the closest to the results of the traditional method when the gray value is 71. The R value porosity curve is closest to the results of the traditional methods when the R value is 55. The G value porosity curve is closest to the results of the traditional methods when the G value is 210. The B value porosity curve is closest to the results of the traditional methods when the B value is 140. According to the above comparison results, the gray value porosity curve is closer to the results of the traditional methods than the RGB value porosity curve is when the gray value is 71.

9

rsos.royalsocietypublishing.org R. Soc. open sci. 0000000

(b) Results of the finite automata image analysis method

This explanation includes information which should be in the methodology section.

In this study, each pixel of the carbonate polarized image was converted into a gray value or RGB value. The Euler number formula used in this study has 9 sets of 20 solutions to the gray value and RGB value of the carbonate polarized image. If we put these 20 solutions into [0,255], there will be 20 image processing thresholds for the gray value or RGB value. The 20 image processing thresholds were all processed using the finite automata. For each polarized image of the carbonate rocks, 20 black and white binary images were obtained using the image processing thresholds of the gray value or RGB value. The porosity was obtained by dividing the number of black pixels by the total number of pixels in the black and white binary images. The porosity of the rock in the polarized image must be greater than 0 and less than 100%. If the porosity is less than 0 or greater than 100%, there is an error in the finite automaton. According to the above method, it was found that when the gray value was 71, the porosity was the closest to the porosity obtained using the traditional method. The black and white binary images obtained using each image processing threshold were listed, and the images shown in Figure 5 were obtained.

Methodology...

Figure 5. The results of the finite automata using the gray value of the images of the carbonate rocks from Baidiao

What is the scale of the images (photo microographies)? What is the size of the area sampled and processed here.

a) Original image; b) The result of the finite automata obtained using a gray value image processing threshold of 11; c) 27; d) 35; e) 50; f) 55; g) 62; h) 71; i) 79; j) 85; k) 95; l) 101; m) 125; n) 135; o) 140; p) 155; q) 160; r) 175; s) 185; t) 195; u) 210.

11

rsos.royalsocietypublishing.org R. Soc. open sci. 0000000

Figure 6. The results of the finite automata using the R value of the images of the carbonate rocks from Baidiao

Same observations as for the Figure 5.

a) Original image; b) The result of the finite automata obtained using a R value image processing threshold of 11; c) 27; d) 35; e) 50; f) 55; g) 62; h) 71; i) 79; j) 85; k) 95; l) 101; m) 125; n) 135; o) 140; p) 155; q) 160; r) 175; s) 185; t) 195; u) 210.

Figure 7. The results of finite automata using the G value of the images of the carbonate rocks from Baidiao

a) Original image; b) The results of the finite automata obtained using a G value image processing threshold of 11; c) 27 ; d) 35; e) 50; f) 55; g) 62; h) 71; i) 79; j) 85; k) 95; l) 101; m) 125; n) 135; o) 140; p) 155; q) 160; r) 175; s) 185; t) 195; u) 210.

Figure 8. The results of the finite automata using the B value of the images of the carbonate rocks from Baidiao

a) Original image; b) The results of the finite automata obtained using a B value image processing threshold of 11; c) 27; d) 35; e) 50; f) 55; g) 62; h) 71; i) 79; j) 85; k) 95; l) B value 101; m) B value 125; n) B value 135; o) B value 140; p) 155; q) 160; r) 175; s) 185; t) 195; u) 210.

Figure 9. Converting the finite automata processing results into rock pores

- a) Finite automata processing results for a gray value of 71; b) Converting a into rock pores; c) Finite automata processing results for a R value of 55; d) Converting c into rock pores; e)

Finite automata processing results for a G value of 210; f) Converting e into rock pores; g) Finite automata processing results for a B value of 140; h) Converting g into rock pores.

(c) 3D data model of the carbonate rocks from Baidiao

In this study, there were 9 sets of solutions for the polarized images of the carbonate rocks from the Baidiao area at [0,255]. In addition, for the initial state when the Euler's number is 0, there are 10 Euler numbers in the Euler number formula for the Baidiao area between [0,255]. All of the Euler numbers, gray values, and RGB values were put into the same coordinate system, and the curve shown in Figure 10 was obtained.

????

Figure 10. Euler number and image processing threshold Cross correlation?

In the gray threshold and Euler number curve, the gray value corresponding to nine Euler numbers divides [0,255] was divided into nine sub-intervals. With reference to the finite automata processing results of the image processing threshold, these nine sub-intervals indicate that as the gray value increases, the porosity of the carbonate rock decreases. According to the results of the Euler's number calculations, the polarized images of the carbonate rocks from the Baidiao area were divided into nine levels according to the porosity of the carbonate rock. The image obtained using the RGB threshold was used as a texture for each surface of the cube.

Why you used 2D images from thin sections to build a cube? It is not clear at all the objective of this procedure. What is the base for this stage.

The simple montage o the images can not produce a realistic representation of the porosity system!

rsos.royalsocietypublishing.org R. Soc. open sci. 0000000

Figure 11. 3D data model of the carbonate rocks from Baidiao

Methodology must be placed in the correct section.

(d) Results of the 16S rDNA analysis

Microbial community genomic DNA was extracted from water samples from the Baidiao area (bd1 and bd2) using TransGen AP221-02 according to the manufacturer's instructions. The DNA extracted was checked using 1% agarose gel, and the DNA's concentration and purity were checked using NanoDrop 2000. The hypervariable region of the bacterial 16S rRNA gene was amplified using an ABIGeneAmp®9700 PCR thermocycler. The purified amplicons were pooled in equimolar and paired-end sequenced using the anIllumina MiSeq PE300 platform/NovaSeqPE250 platform (Illumina, San Diego, USA) according to the standard protocols provided by Majorbio Bio-Pharm Technology Co. Ltd. (Shanghai, China). **Based on the results obtained using the above research methods, the water samples collected from the Baidiao area do not contain nitrifying bacteria.**

This figure must be explained in the text above. This figure is not mentioned in the text above.

17

rsos.royalsocietypublishing.org R. Soc. open sci. 0000000

Figure 12. Community heat map analysis on the phylum level

A1: Gemmatimonadetes; A2: Acidobacteria; A3: Euryarchaeota; A4: Kiritimatiellaeota; A5: Nanoarchaeaeota; A6: Fibrobacteres; A7: Dependuntiae; A8: Omnitrophicaeota; A9: Epsilonbacteraeota; A10: Lentisphaerae; A11: Armatimonadetes; A12: Fusobacteria; A13: Hydrogenedentes; A14: Elusimicrobia; A15: DeinococcusThermus; A16: Patescibacteria; A17: Chlamydiae; A18: Firmicutes; A19: Chloroflexi; A20: Planctomycetes; A21: unclassified_knorank_dBacteria; A22: Verrucomicrobia; A23: Bacteroidetes; A24: Actinobacteria; A25: Spirochaetes; A26: Cyanobacteria; and A27: Proteobacteria.

4. Discussion and analysis

(a) Comparison with the research results obtained using traditional methods

The carbonate porosity obtained using the image threshold processing method and the Euler's number method and the carbonate porosity obtained using the traditional method were put in the same coordinate system and were compared using a histogram (Figure 13). As can be seen from Figure 13, the carbonate porosities of T_{2y}^6 and T_{2b} obtained using the three methods are relatively close. The differences in the carbonate porosities obtained using the three methods of $T_{2y}^{5-1}-2$ are also acceptable. Based on the five samples, the accuracy rate exceeds 40%. The accuracy rate is higher than that of direct judgment using the human eye. This is higher than the accuracy rate of human judgment in the engineering department in Baidiao (about 20%).

which methods (three?)?

Rock porosity is not defined through visual description of thin sections. It provides only approximation, which is not used as a precise information in the characterization of reservoir rocks.

Except for T_{2y}^4 , the porosities obtained using the image threshold processing method and the Euler number method for all of the samples were relatively close. This shows that the image threshold processing method and the Euler number method do not conflict with the image processing algorithms applied to the study of carbonate porosity. **The number of polarized rock images used in this study is relatively small.** If the number of polarized rock images used in the study were increased, there is still room for improvement in the image processing algorithm, and there should be room for improvement of the accuracy.

????

You should discuss the results in contrast with other works which performed similar analyses. Conclusions should be kept for the adequate section.

Use the caption and the text above to explain the information in this graph.

Figure 13. Comparison of the carbonate porosities obtained using the three methods

A) Porosity obtained using the image threshold processing method, B) porosity obtained using the traditional methods, C) porosity obtained using the Euler number method

(b) Using Euler numbers to build a carbonate pore model

In this study, the Euler number of the polarized image of the carbonate rock was put in a three-dimensional coordinate system to determine the intersection points, and these intersection points were represented by cubes to establish a three-dimensional pore model of the carbonate rock.

In order to control the amount of calculations required, a 3×8 matrix was used to organize the Euler numbers, and the number of intersection points was kept below eight. These cubes represent the pores in the carbonate rocks. Since the results of the image threshold method

??????

Methodology

and the Euler number method are too different for T_{2y}^4 , its Euler number was not used in the modeling. As can be seen from Figure 14, the cubes constructed using the Euler number intersection points are greater in T_{2y}^6 , T_{2y}^{5-1-1} , and T_{2b} in the horizontal direction and are greater in T_{2y}^{5-1-2} in the vertical direction. Therefore, in T_{2y}^6 , T_{2y}^{5-1-1} , and T_{2b} , the uniaxial compressive strength in the vertical direction is greater than in the horizontal direction. In T_{2y}^{5-1-2} , the uniaxial compressive strength is greater in the horizontal direction than in the vertical direction.

This information is completely disconnected from the discussion about the porosity, and the correlation does not make any sense in the way it is described.

Figure 14. Using Euler numbers to build a carbonate pore model

(c) Causes of the rock pores in the carbonate rocks in the Baidiao area

This assumption need to be addressed with evidence.

The pores in the carbonate rocks in the Baidiao area should be caused by the water from the Yalong River. The microorganisms in the Yalong River water and the microorganisms encountered when the water penetrates the surface soil enter the pores in the carbonate rocks in the Baidiao area. These microorganisms contain nitrifying bacteria. In order to maintain their own survival, these nitrifying bacteria will change the amount of hydrogen ions in the karst water, affecting the karst process of the carbonate rocks in the Baidiao area. There is feldspar

and pyrite in the carbonate rocks in the Baidiao area. The combination of the feldspar and pyrite and the nitrifying bacteria will change the karst process of the carbonate formation. In summary, the nitrifying bacteria in the carbonate strata in the Baidiao area affect the karst process of the carbonate rocks and expand the pores in the carbonate rocks. Through the 16S rDNA analysis of the river water from the Baidiao area, it was found that there are nitrifying bacteria in the local Yalong River water.

This information is contrary to what is mentioned in the lines 56 to 58. What is the correct result?

The simplistic assumptions made are not sufficient to provide any conclusion about the relation between bacteria activity and the origin of the porosity

(d) The advantages and disadvantages of using finite automata in carbonate pore research

The advantages of using finite automata to study the pores in carbonate rocks are obvious. The processing results of the polarized images of the carbonate rocks obtained using the finite automata are very intuitive. Even people who have never learned image processing can perform pore analysis of carbonate rocks based on the results obtained from the image processing using the finite automata. Since the pixels satisfy the condition that the finite automata are finite, the quantitative calculation of the porosity can be easily performed. The calculated results can be compared with the porosity results obtained using other research methods. The porosity results obtained using other research methods can also help improve the finite automata algorithm. The application of the finite automata to the study of carbonate pores also has obvious disadvantages. The premise of the finite automata is that the researcher must be familiar with the formal language. However, many researchers of carbonate rocks are not familiar with the formal language. Finite automata require correct image processing thresholds, but many researchers who are proficient in the formal language are not familiar with carbonate images. Thus, this requires researchers to pay attention to both the formal language and carbonate learning.

You need to use references and a much better comparison between the results and similar research found in the literature. The discussion presented here is totally incipient and does not allow the reader to evaluate any contribution.

(e) Can finite automata be used to study carbonate pores in other regions?

The Baidiao area in the Jinping area is surrounded by the Yalong River basin and has a wide distribution of carbonate rocks. In the Jinping area, excluding the Baidiao area, there are many areas where studies have been carried out on the karst development, uniaxial compressive strength, and rock permeability of carbonate rocks. These research results can be used in the study of the image processing threshold of the finite automata. Thus, the research method presented in this paper can be used in the Jinping area. Due to engineering construction, there are a large number of processed carbonate rock specimens for uniaxial compressive strength analysis and carbonate rock lithology analysis slides from the Jinping area. Therefore, the cost of processing carbonate rock uniaxial compressive strength specimens and carbonate rock lithology analysis slides is not high in the Jinping area. This is also one of the reasons why the research method presented in this paper can be applied in the Jinping area. If we want to apply the research method presented in this paper to carbonate rock areas outside of Jinping, it is best that the following conditions be met. Researchers have received formal language training in GeoAgent and finite automata. Relatively long-term research has been conducted on the uniaxial compressive strength, karst development speed, and rock permeability of the local carbonate rock formation. The research data are sufficient to support the establishment of the image processing threshold. The cost of local carbonate image acquisition is not high.

5. Conclusions

Through the use of finite automata in the Baidiao area to study polarized images of carbonate rocks, the following conclusions have been drawn.

- 1) It is feasible to use finite automata to study the porosity of the carbonate rocks in the Baidiao area. Compared with the porosity obtained using traditional research methods, the accuracy of the proposed method can reach more than 40%.
- 2) The accuracy of using the finite automata to study the pores in polarized images of the carbonate rocks in the Baidiao area is higher than that of the empirical judgment by researchers.
- 3) The use of the finite automata for pore research of the polarized images of the carbonate rocks in the Baidiao area was mainly established using the image processing threshold, which is mainly composed of the gray value, RGB value, or Euler number.
- 4) If the gray value is used to determine the image processing threshold of the finite automaton, the method of gradually approximating the results of the traditional research methods is more reliable.
- 5) In the Baidiao area, if the Euler number is used to determine the image processing threshold of the finite automata, the final Euler number must be converted to a gray value or RGB value, which is then used as the image processing threshold of the finite automata.
- 6) The results of using the finite automata to study the pores in the polarized images of the carbonate rocks in the Baidiao area show that when the gray value is as the image processing threshold the results are closer to the results obtained using the traditional methods than the Euler number results are.
- 7) The research results for the Baidiao area show that polarized images of carbonate rocks can be used to construct a 3D model of carbonate pores. The carbonate rock pore model constructed using the Euler number can be used to help analyze the uniaxial compressive strength of the rock.
- 8) The research results for the Baidiao area show that when using the finite automata to study the pores in carbonate rocks, the gray values are more suitable as image processing thresholds than the RGB values.
- 9) The research results for the Baidiao area show that the Euler number is a very important research method when using the finite automata to study the pores in carbonate rocks.
- 10) The 16S rDNA test results of the water samples from the Baidiao area showed that the local Yalong River water contains nitrifying bacteria. It is possible that the expansion of the carbonate pores in the Baidiao area originated from the microorganisms in the karst water.
- 11) The research method presented in this paper can also be used for the other parts of the Jinping area, i.e., outside of the Baidiao area.
- 12) In other regions where porous carbonate rocks are distributed, if there is a large amount of data on the rock uniaxial compressive strength, karst development speed, and rock permeability and the cost of making rock specimens and polarized glass slides is not high, the research method presented in this paper can also be used in these regions.

Ethics. There is no ethics text in this paper.

Data Accessibility. Accession to cite for these SRA data: PRJNA703089.

Authors' Contributions. Fig 9 was drawn by Xi Ye, Fig 12 was drawn by Zhiyi Qing.

Acknowledgements. We thank LetPub for its linguistic assistance during the preparation of this manuscript.

Disclaimer. There is nodisclaimer text in this paper.

References

1. Zhang jiqun, Hu changjun HdCJLxLh. 2015 image analysis method of pore structure and its application in rock image.. *Well logging technology* **39**, 550–554.
2. Sun wenfeng, Li wei DzYtLyYs. 2017 image analysis method of pore structure and its application in rock image.. *lithologic reservoirs* **29**, 125–130.
3. Zhang jiazheng, chen songling cy. 2008 Carbonate dagenesis and porosity evolution in adjacent arch of Nanpu depression.. *journal of oil and gas technology* **30**, 161–165.
4. Peiqing Lian, Wenbin Gao XTDFWHZYL. 2020 Workflow for pore-type classification of carbonate reservoirs based on CT scanned images.. *Oil & gas geology* **4**, 852–861.
5. Qiangfu Kong, Cai Yang HLCGJD. 2020 A lithology recognition method based on multi-resolution graph-based clustering and K-Nearest Neighbor:A case study from the Leikoupo Formation carbonate reservoirs in western Sichuan Basin.. *Oil & gas geology* **4**, 884–890.
6. WANG Lu, YANG Shenglai LYWYMZHWQK. 2017 Visual experimental investigation of gas-water two phase micro seepage mechanisms in fracture-cavity carbonate reservoirs.. *Petroleum Science Bulletin* **2**, 364–376.
7. CHEN Yulin, ZENG Yan DYWQ. 2018 Pore structure characteristics and reservoir classification of dolomite reservoirs in fourth member of Leikoupo Formation, Longmen Mountain front, western Sichuan Basin.. *Petroleum Geology & Experiment* **40**, 621–631.
8. Xie shuyun, he zhiliang qyfhztzd. 2015 multifractality of 3D pore structures of carbonate rocks based on CT images.. *journal of geology* **39**, 46–54.
9. Wang chenchen, yao jun yywxjgy. 2013 structure characteristics analysis of carbonate dual pore digital rock.. *journal of china university of petroleum* **37**, 71–74.
10. Shou jianfeng, she min sa. 2016 experimental simulation of dissolution effect of carbonate rock under deep burial condition.. *bulletin of mineralogy,petrology and geochemistry* **35**, 860–867.
11. WU Guo-ming, LI Xi-zhao GSsea. 2017 Exploring the best threshold of binary CT image of carbonate rock based on fractal theory.. *Oil Geophysical Prospecting* **52**, 1025–1032.
12. Wang Dengke, Zhang Ping PHea. 2018 Experimental research on cracking process of coal under temperature variation with industrial micro-CT.. *Chinese Journal of Rock Mechanics and Engineering* **37**, 2243–2252.
13. Chai hua, li ning xsea. 2012 high-resolution rock structure image processing method and its applications in carbonate reservoir evaluation.. *acta petrolei sinica* **33**, 154–159.
14. Qin yujuan, zhang tianfu hyzy. 2013 the significance of a back-scattered electron image(of EPMA) in micro-area analyses of carbonate rocks.. *journal of chinese electron microscopy society* **32**, 479–484.
15. Wang fenge zc. 2009 realization of damage fractal dimensions of rock SEM image in the matlab enviroment.. *ship electronic engineering* **29**, 144–146.
16. Ye runqing, niu ruiqing zl. 2011 mineral features extraction and analysis based on multiresolution segmentation of petrographic images.. *journal of jilin university(earth science edition)* **41**, 1253–1261.
17. Cheng guojian, yang jing hqly. 2013 rock image classification recognition based on probabilistic neural networks.. *science technology and engineering* **13**, 9231–9235.
18. Wang weixing, yu xin lj. 2009 image enhancement for rock fractures based on fractional differential.. *journal of computer applications* **29**, 3015–3017.
19. Dang fuxing, fang hongbin zf. 2002 techniques of geological remote sensing information extraction from CBERS-1 CCD data for mineral exploration.. *remote sensing for land & resources* **4**, 51–66.
20. Tannaz Pak, Ian B. Butler SGMIJvDKSS. 2015 Droplet fragmentation:3D imaging of a previously unidentified pore-scale process during multiphase flow in porous media.. *PNAS* **112**, 1947–1952.
21. Seyyedi, M. MHVMa. 2020 Pore Structure Changes Occur During CO₂ Injection into Carbonate Reservoirs.. *Sci Rep* **10**, 3624.
22. Ghiasi-Freez J, Honarmand-Fard S, Ziaii M. 2014 The Automated Dunham Classification of Carbonate Rocks Through Image Processing and an Intelligent Model.. *Petroleum Science and Technology* **32**, 100–107.

23. Goral J. DM. 2020 Nanofabrication of synthetic nanoporous geomaterials: from nanoscale-resolution 3D imaging to nano-3D-printed digital (shale) rock. *Sci Rep* **10**, 21256.
24. Kotz Fea. 2019 Fabrication of arbitrary three-dimensional suspended hollow microstructures in transparent fused silica glass.. *Nat. Commun.* **10**, 1–7.
25. Ishutov S., Hasiuk FJJDas. 2018 Using resin-based 3D printing to build geometrically accurate proxies of porous sedimentary rocks.. *Groundwater* **56**, 482.
26. Golreihan A, Steuwe C WLDafyVJea. 2018 Improving preservation state assessment of carbonate microfossils in paleontological research using label-free stimulated Raman imaging.. *plos one* **13**, e0199695.
27. Goral, J. DM. 2020 Nanofabrication of synthetic nanoporous geomaterials: from nanoscale-resolution 3D imaging to nano-3D-printed digital (shale) rock.. *Sci Rep* **10**, 21596.
28. Berg, C. F. LOBH. 2017 Industrial applications of digital rock technology.. *J. Petrol. Sci. Eng* **157**, 131.
29. Ali, M.Y. WASMea. 2020 Geophysical imaging of ophiolite structure in the United Arab Emirates.. *Nat Commun* **11**, 2671.
30. Lanari Pea. 2014 XMapTools a Matlab©-based graphic user interface for microprobe quantified image processing.. *Comput. Geosci* **62**, 227–240.
31. Saeed Yarmohammadi, David A.Wood AK. 2020 Reservoir microfacies analysis exploiting microscopic image processing and classification algorithms applied to carbonate and sandstone reservoirs.. *Marine and Petroleum Geology* **121**, 104609.
32. KURZ TH. 2011 Hyperspectral image analysis of different carbonate lithologies (limestone, karst and hydrothermal dolomites): the Pozalagua Quarry case study (Cantabria, North-west Spain). *Sedimentology* **59**, 623–645.
33. Fusi N MMJ. 2013 Mercury porosimetry as a tool for improving quality of micro-CT images in low porosity carbonate rocks. *Engineering Geology* **166**, 272–282.
34. Harris. K. 2013 A case of lanthanum carbonate ingestion thought to be phlebosclerotic colitis on CT imaging and abdominal radiograph. *Radiography* **23**, e23–e26.
35. Maheshwari P. 2013 Comparison of Carbonate HCl Acidizing Experiments with 3D Simulations. *SPE Production & Operations* **28**, 402–413.
36. Munoz H. 2016 Pre-Peak and Post-Peak Rock Strain Characteristics During Uniaxial Compression by 3D Digital Image Correlation. *Rock Mechanics and Rock Engineering* **49**, 2541–2554.
37. Sharafisafa M. 2020 Experimental Investigation of Dynamic Fracture Patterns of 3D Printed Rock-like Material Under Impact with Digital Image Correlation. *Rock Mechanics and Rock Engineering* **53**, 3589–3607.
38. Kim H. 2020 Stress Estimation through Deep Rock Core Diametrical Deformation and Joint Roughness Assessment Using X-ray CT Imaging. *Sensors* **20**, 6802.
39. Schepp, L.L. ABBMea. 2020 Digital rock physics and laboratory considerations on a high-porosity volcanic rock. *Sci Rep* **10**, 5840.
40. Saenger EHea. 2016 Analysis of high-resolution X-ray computed tomography images of Bentheim sandstone under elevated confining pressures.. *Geophysical Prospecting* **64**, 848–859.
41. Osorno, M. UDROESH. 2015 Finite difference calculations of permeability in large domains in a wide porosity range. *Archive of Applied Mechanics* **85**, 1043–1054.

Appendix B

Response to Reviewer comments :

to all reviewers:

This paper is not authorized to publish geological data in Baidiao area. However, the reviewers all asked for the burial depth and stratum thickness of the rock specimens. For this reason, this paper chooses to expand the research area to Jingfengqiao-Baidiao area. The rock specimen in this article comes from a borehole (No: Qx404). The coordinates and precise map of Qx404 cannot be authorized. In this paper, all the rock samples obtained by Qx404 were re-experimented according to the method of Baidiao area. To ensure the repeatability of this article, polarized image of rock samples obtained by Qx404 are submitted. Since the BaiDiao area is the area in Jinping where Denitrifying bacteria exist in the water samples currently collected, the 16S RDNA data of the BaiDiao area is retained.

Reviewer: 1

cite: Experimental Materials: Please give the depth of each sample with its formation.

reply: Modifications have been made in "Experimental materials": "In this study, Three rock samples at the depths of 69.80-71.01m were collected from borehole(No: Qx404) in Jingfengqiao area, 2 water samples (BD1 and BD2) were collected in the Hebian area, Baidiao, and water samples of Baidiao were transported to Chongqing within 27 hours. Rock samples of Baidiao were transported to Chongqing within 480 hours. The rock samples were collected from T_{2z} formations. The thickness of the T_{2z} formation in Jingfengqiao area is about 150m-700m. The rock sample numbered JP12 is griotte, with a $CaCO_3$ content of about 95%. Therefore JP12 is a very pure carbonate rock. This paper uses JP12 rock samples to process two rock slides. The porosity of the JP12 rock samples obtained by the TCRM method is 0.19%."

cite: • Traditional Method of calculating rock porosity: Please include oven temperature for drying process.

reply: The oven temperature has been indicated in the text. 60°C

cite: Please remove Figure 3, Figures 3 and 4 have the same data.

reply:ok

cite: • In Figure 4, (new Figure 3) colors of the curves should match with RGB and gray colors, please include a legend as well.

reply:OK, The new figure has been modified in accordance with this recommendation.

cite: • Please do not use "we" in your text. Rephrase all those sentences in passive voice.

reply: So the sentences at the beginning of "we" have been rewritten and polished. A total of 11 changes were made before polishing.

cite: Please explain Figure 12 more: what is phylum level? why are you looking at this phylum level? What is the differences between red and blue regions? Which of those bacteria are nitrifying bacteria in this list in Figure 12?

reply: The representative sequence of each OTU is compared with the known sequence, and each OTU is traced back to its species source, divided into: Kingdom, Phylum, Class, Order, Family, and Genus, species. Add the species abundances of the same level to sort out the species abundance value of each level. For example, adding the species abundances of the same family in the family level to form a family level species abundance value; adding the species abundances of the same phylum in the phylum level to form a phylum level species abundance value. Therefore, the abundance value of phylum level is an important research method for 16S rDNA^{1,2}. The following text is added to the body :" **This paper uses OTU analysis to test the microbial diversity and the abundance of different microorganisms in the water samples of Baidiao area. Old Figure 12(new fig8) is the community heatmap analysis on phylum level. Each column represents 1 sample. Each row represents phylum. The color blocks represent species abundance values. As the legend shows, the more red the color, the higher the abundance value. The bluer the color, the lower the abundance. From Figure 12, it can be seen that the abundance of unclassified_k_norank_d_Bacteria(A21) is higher in both samples. Therefore, there are denitrifying bacteria in the water samples in the Baidiao area.**"

reference:

1YAN Yuan et al, Microbial Community Characteristics of a Completely Autotrophic Nitrogen Removal Over Nitrite (CANON) System Based on High-throughput Sequencing Technology, Journal of Beijing University of Technology,10, 1485-1492(2015)

2 C T, Swamy & Devaraja, Gayathri. (2021). High throughput sequencing study of foliose lichen-associated bacterial communities from India. Molecular Biology Reports. 48. 10.1007/s11033-021-06272-6.

cite: • Figures 1, 6-9, 11, 12 were not mentioned in the text. Please mention them in the manuscript.

reply: Add the following description to Figure 1:" **In actual engineering applications, engineers in Jinping area often use the same polarized image of the rock as a texture to construct a 3D rock model. Engineers use experience to judge the water permeability of the 3D rock model. When engineers' judgments conflict, the opinions of older engineers are often adopted. This paper attempts to improve the research method and make it more convincing. The water permeability of a rock cube is different on different sides. Using only one picture as the texture of the rock cube is not enough to reflect the actual water permeability of the rock. So as shown in Figure 1,the classified image was used as the texture of the 3D cube and was pasted it onto each surface of the cube, and then, 3D models of the carbonate rock could be obtained.**"

Add the following description to Figure 4-7(old 6-9):"**From Figures 4, 5, 6, and 7, the processing effect of gray threshold (Fig 4) shows that the difference between different gray thresholds is relatively small, unlike the processing effect of RGB threshold (Fig 5,6,7), it shows The difference between different thresholds is relatively large.Figures 3, 4, 5, 6, and 7 all show that the porosity obtained by the gray threshold is closer to the actual value measured by the TCRM method than the RGB threshold.**"

Add the following description to Figure 8 (old FIG 12) : " From Figure 8, it can be seen that the abundance of unclassified_k_norank_d_Bacteria(A21) is higher in both samples. "

cite: • Under “Results of 16S rDNS analysis” : You are mentioning that” the water samples collected from the Baidiao area Do NOT contain nitrifying bacteria”, but under “(c) Causes of the rock pores in the carbonate rocks in the Baidiao area” , you are saying: “Through the 16Sr DNA analysis of the river water from the Baidiao area, it was found that there are nitrifying bacteria in the local Yalong River water” . Which one is true or is it typo mistake?

reply: It can be seen from Figure9 (old FIG 12) that the water samples in the Baidiao area have denitrifying bacteria but no nitrifying bacteria. Denitrifying bacteria are "反硝化菌" in Chinese, and nitrifying bacteria are "硝化菌" in Chinese. The Chinese of the two are a bit similar. English translators translated both denitrifying bacteria and nitrifying bacteria into nitrifying bacteria.

“Results of 16S rDNS analysis” is right, the water samples collected from the Baidiao area Do NOT contain nitrifying bacteria, There are only denitrifying bacteria and no nitrifying bacteria in the water samples in the Baidiao area. " nitrifying bacteria " in “(c) Causes of the rock pores in the carbonate rocks in the Baidiao area” should be replased by denitrifying bacteria. The English polisher did not see the problem. The authors did not see this problem when they checked. This is a translation error. The nitrifying bacteria and denitrifying bacteria in the article have been checked and corrected.

cite:• Please edit your references based on Journal’ s referencing criteria and be consistent with one criterion.

reply: The references are all organized in the following format provided by the template:

@article {BF,

AUTHOR = {Benjamin, T. B. and Feir, J. E.},

TITLE = {The disintegration of wave trains on deep water. {P}art 1. {T}heory},

JOURNAL = {J. Fluid Mech.},

FJOURNAL = {Journal of Fluid Mechanics},

VOLUME = {27},

YEAR = {1967},

NUMBER = {3},

PAGES = {417--437},

ISSN = {0022-1120},

}

cite:• Please use last names in citations and references, not the first names.

Thanks.

reply: According to this recommendation, this article has made 23changes. Full names are used for authors whose family names are difficult to identify.

Reviewer: 2

Comments to the Author(s)

cite:1. In the section of Introduction, the methods should be classified by the essence and some latest relevant references are lacking.

reply: The introduction of this thesis is about the current status of Chinese research in the subject first, followed by the current status of foreign research. According to this recommendation, the current research status of this subject in China and foreign research in the introduction of this paper are divided into four categories: the application of hardware equipment in the study of rock pores in China, the application of software in the study of rock pores in China, and the application of hardware equipment in foreign countries and the application of software in foreign rock pore research. Each category has added 3 research results since January 1, 2020 as references, and a total of 12 references have been added.

CITE:2. In Figure 4., you should give specific legends, otherwise, it's easy to fall into confusion.

REPLY: According to the recommendation of Reviewer 1, OLD FIG3 was deleted, and OLD FIG4 is now FIG3. Your recommendation is consistent with the recommendation of Reviewer 1. A legend has been added to FIG3.

Figure 3. Image processing threshold curve comparison

CITE:3. In the first paragraph of page 10, the author keeps mentioning the results of traditional methods, but the paper does not find specific values or charts for comparison.

REPLY: The traditional carbonate research method (TCRM) is relatively mature and can be represented by the following figure:

The first author faces the pressure of the number of scientific research papers, so the traditional carbonate research method (TCRM) is the research content of another paper, so the above picture can not be used in this article, and is only for reviewers' reference .

In order to help the reviewers better review this article, the author gives the rock porosity determined by traditional carbonate research methods in " Experimental materials ".

CITE: 4. The formula in the paper needs to be explained to the variables.

REPLY: The text of 2.1 added before polishing is: R, G, B are the RGB value of the pixel; n is the sequence number of the pixel; E_n is the Euler number of the pixel calculated by the RGB value and sequence number of the pixel; Gray is the pixel gray value calculated by the pixel RGB value;

The text of 2.2 added before polishing is: k is a finite state set ; $Gray_{\{(i,j)\}}, R_{\{(i,j)\}}, G_{\{(i,j)\}}, B_{\{(i,j)\}}, E_{n_{\{(i,j)\}}}$ are finite state of k ; Σ is map list; $T_{\{1(i,j)\}}, T_{\{2(i,j)\}}, T_{\{3(i,j)\}}, T_{\{4(i,j)\}}, T_{\{5(i,j)\}}$ are maps of Σ ; i, j is the row and column number of the pixel; f is the result set of Σ ; $q_{\{(i,j)\}}$ is the mapping result of Σ ; $A_{\{(i,j)\}}$ is new value of color value; z is the result state set.

CITE: 5. In the section of Comparison with the research results obtained using traditional methods, judging from the results in Figure 13, I think there may be some errors in your description. For example, isn't 0.84 in T42y closer to 0.87? This is contrary to the description in the article.

reply:

Figure 13. Comparison of the carbonate porosities obtained using the three methods

A) Porosity obtained using the image threshold processing method, B) porosity obtained using the traditional methods, C) porosity obtained using the Euler number method

This article was asked about drilling information when it was revised, so the sample collection area was changed. The old fig 13 has been modified to the Jingfengqiao picture(new fig 9 and 10). In order to answer this question, the old fig 13 is specially inserted to make the following explanation:0.84 is porosity obtained using the image threshold processing method;0.92 is porosity obtained using the traditional methods; The accuracy of T_{2y}^4 is lower than that of T_{2b} .In old fig13, T_{2y}^6 、 T_{2y}^4 、 T_{2b} is close to porosity obtained using the traditional methods.0.87 is porosity obtained using the euler number method; The porosity obtained using the euler number method should not be compared with the porosity obtained using the image threshold processing method. To avoid misunderstandings,the porosity of the two method which is closest to the traditional method should be selected for comparison with the porosity of the traditional method.So in new fig 10, the porosity of the two method which is closest to the traditional method was selected for comparison with the porosity of the traditional method.

Figure 9: Carbonate pore map based on image threshold analysis

Figure 10: Comparison of carbonate pore value obtained by image threshold analysis method and TCRM(A:carbonate pore value obtained by image threshold analysis method;B:carbonate pore value obtained by TCRM)

I think the error in this opinion probably refers to this:

Since the results of the image threshold method

and the Euler number method are too different for T_{2y}^4 , its Euler number was not used in

Yes, you are right, here should be T_{2y}^6 , In old fig13, T_{2y}^6 should be T_{2y}^4 , After this article was written in Chinese, My student (the original second author and the third author) translated the paper into English, and then polished by a polishing company. My student made an error when copying and pasting the formation number. I didn't find it when I checked. The translation company should not be blamed. Since there are not as many samples in the JINGFENGQIAO area as baidiao, the new Figures 9、 10 and 11 are redrawn.

Figure 11. Using Euler numbers to build a carbonate pore model

cite:6. In the conclusion part of this paper, the author has drawn many useful conclusions. In this paper, the carbonate porosity was obtained by image threshold processing and compared with the traditional method, the accuracy was improved to 40%. However, the final threshold still needs to be determined by approaching the traditional method, which also requires a lot of labor. Should you consider skipping this step and directly determining the threshold?

reply: Indeed, it is very troublesome to conduct traditional laboratory technology research after image processing research. This article originally used the existing results of traditional carbonate research methods to construct a traditional research method database, input the porosity, uniaxial compressive strength and other indicators of traditional carbonate rock formations into the database. The porosity obtained by image processing methods is compared with the value in the database of the same strata in the neighboring area or same area. This can greatly improve research efficiency and speed. and this part is deleted later. I hope I can make this part public in the future.

cite:7. It is noted that your manuscript needs careful editing by someone with expertise in technical English editing paying particular attention to English grammar, spelling, and sentence structure, for helping the reader better understand the goals and results of this study.

reply: This article was first written in Chinese. After the officials had revised it, my students translated the Chinese into English, and then asked the translation company to polish it in English. Before submitting this article again, two foreign language students were asked to modify it twice, and let LETPUB company to polish it again.

cite:Reviewer: 3

Comments to the Author(s)

See the observations in the annotated PDF file of the manuscript

("RSOS-210426_Proof_hi_rev.pdf").

reply: The file I received is 41163565_File000005_1046658560.pdf. To prevent errors in understanding, I took screenshots of the review comments in the paper to avoid errors.

cite:

Subject Areas:

Karst, FA, rock

Keywords:

carbonate, FA, porosity

Author for correspondence:

Honghai Kuang

e-mail: hhkuang@swu.edu.cn

In this study, the gray value, RGB value, and Euler number of the polarized images of the carbonate rocks in the Baidiao area are used to construct a finite automaton and the finite automaton is used to perform black and white binary processing on the polarized images of the carbonate rocks. The porosity of the carbonate rock is calculated based on the

The text needs an extensive revision of the English language -- (see the annotations)

You should mention that the study is based on the processing of computerized microtomography images of rock samples.

reply: This part of the resubmitted version has been rewritten and has been polished by a professional company:

The study is based on the processing of computerized microtomography images of rock samples. In this study, a finite automaton was constructed by the gray value, RGB value, and Euler number of the polarized images of the carbonate rocks in the Jingfengqiao-Baidiao area. The finite automaton is used to perform black and white binary processing on the polarized images of the carbonate rocks.

The above content is not the final text, it is finalized after polishing.

cite:

Experimental methods for porosity determination?

obtained via traditional research methods. When the two porosities are close, the image processing threshold of the finite automata is considered to be credible. Based on the finite automata established using the image processing threshold, the black and white binary images of the polarized images of the carbonate rocks are used to establish a rock porosity

reply: The Chinese manuscript of this sentence is: 传统碳酸盐岩研究方法, The correct translation is: traditional carbonate research methods (TCRM), Sorry I did not find this error.

TCRM is a very common carbonate research method in China, including porosity, uniaxial compressive strength and other aspects.

CITE:

Which classification was applied?

Carbonate rocks are used to establish a rock pore map. The polarized images of the carbonate rocks are classified according to their RGB values using the finite automata for porosity classification, and the

REPLY: As shown in Figure 1 and 4-7, all carbonate rock polarized images are divided by different image threshold, representing different degrees of water permeability.

cite:

This method is not applied for porosity measurements in rock samples by the industry. The porosity is usually measured through controlled liquid or gas injection in the rocks, and density values can be used to calibrate porosity essays.

1. Introduction

The study of the porosity of carbonate rocks is important to karst investigations. The cost of the widely used dry and wet weighing method to study rock porosity is relatively high. Therefore, it is necessary to use computer image analysis to study the porosity of carbonate rocks.

© 2014 The Authors. Published by the Royal Society under the terms of the Creative Commons Attribution License

reply: In the Jinping porous carbonate area, the dry-wet weighing method is a relatively frequently used method for studying the porosity of carbonate rocks. Because ovens are not available everywhere, engineers in Jinping area generally use gasoline to soak the rocks. Many engineers in Jinping area use this method. Many engineering project reports in Jinping area have research data of this method. The reviewer's opinions must be obeyed, and this sentence has been deleted in the resubmitted version.

cite:

To study the porosity of carbonate rocks using computer image analysis, we must first choose an appropriate algorithm. If the computer image analysis algorithm is not accurate, the reliability of the results of the carbonate porosity study will be questionable. It is best to use traditional research methods to verify the accuracy of the carbonate porosity results obtained using computer image analysis. The expansion of carbonate pores is sometimes related

Confusing sentence

reply: The sentence with the red line has been deleted. The yellow text was changed to: This study tried to use the traditional carbonate research method (TCRM) to verify the accuracy of the carbonate porosity results obtained using computer image analysis. The above content is not the final text, it is finalized after polishing.

cite:

It is a simplistic assumption. The occurrence of bacteria in the pore water does not automatically mean that the dissolution was to microorganisms in karst water. In order to determine whether or not there are microorganisms that affect the karst process in karst water, researchers only need to perform a 16S rDNA test. Jiqun et al. discussed the image analysis method of analyzing the pore structure and its application to rock images[1]. They illustrated that the black and white binarization algorithm based on the gray value is very important in the computer image analysis of rock pores. Wenfeng et al. found a new approach to the characterization of the pore structure of shale[2]. Their results helped improve the algorithm used in this study. Jiazheng et al. conducted research on the diagenesis and pore evolution of carbonate rocks in the Nanpu area[3]. They found that the effect of microorganisms on rock pores cannot be ignored. Lian et al. proposed a new

reply: This paper mainly focuses on the application of computer image analysis technology in the

study of carbonate pores. The expansion effect of karst water and soil microorganisms on carbonate pores in Jinping porous area is the research content of other papers. According to the existing research results (which are waiting for publication), the expansion of the pores of carbonate rocks in the Jinping porous area by microorganisms in karst water is mainly realized by the following methods:

The above formula is the content of my other papers, and is only for reviewers' reference, not for use in this paper. It mainly explains the influence of microorganisms on karst water in Jinping area.

cite:

gas-water microseepage mechanisms in fracture-cavity carbonate reservoir of **great help to the traditional research methods used in this study.**
~~what traditional method?~~ pore structure characteristics and reservoir classification of the dolomite member of the Leikouno Formation, at the foot of Longmen Mountain.
 reply: The newly submitted version changed to "traditional carbonate research methods(TCRM)".
 According to another reviewer's opinion, the location of this reference has been adjusted.

cite:

~~imaging of carbonate rocks based on fractal theory[11]. They confirmed that carbonate porosity can be obtained through image binarization operation.~~ Dengke et al. conducted experimental research on the cracking process of coal under temperature variations using industrial micro-CT[12]. ~~They demonstrated that microscopic techniques can be used for rock pore research.~~

reply: These have been deleted.

cite:

~~in the micro-area analysis of carbonate rocks~~[14]. They showed that imaging and **micro-area analysis of carbonate rocks** is an effective method for carbonate rock research. Fengge et al. ~~11127777~~

reply: The concept of "micro-area analysis of carbonate rocks" comes from the following:

The significance of a back-scattered electron image (of EPMA) in micro-area analyses of carbonate rocks

QIN Yu-juan^{1,2}, ZHANG Tian-fu^{1,2}, HU Yuan-yuan^{1,2}, ZHU Yin^{1,2}

(1. PetroChina Hangzhou Research Institute of Geology, Hangzhou Zhejiang 310023;

2. Key laboratory of Carbonates Reservoirs, CNPC, Hangzhou Zhejiang 310023, China)

Abstract: Though the compositions of carbonate rock are simple, its structure is rather complicated, so micro probe analysis is very important for the investigation. However, the conventional apparatus, such as the optical microscope, has no further capability to improve the studying precision of the micro-area analyses of carbonate rocks. But the advanced apparatus like EPMA has the ability to enhance greatly both the research precision and profundity. It can magnify the object by thousands of times, and even one-hundred-thousand times. The images acquired can distinctly show microscale to nanoscale microtextures. At the same time, the elements can be analyzed correspondingly. By connecting the microtextures with chemical compositions of a sample, EPMA performs *in situ* and credible micro-area analyses accurately. In this paper, the author emphasizes a sort of foremost image in EPMA, namely a Back-Scattered Electron Image (BSEI), which plays a leading role in the micro-area analyses of carbonate rocks. Besides the significance in studying microtextures and positioning precisely for element analyzing, the significance in identifying minerals is also crucial.

Keywords: BSEI EPMA; microtexture micro-area analyses; carbonate rocks

According to this article, "micro-area" refers to a very small area in a carbonate microscopic image (the original text does not give an accurate area)

cite:

of Euler numbers must be emphasized in the study of carbonate pores. Guojian et al. studied rock image classification recognition based on probabilistic neural networks[17]. They illustrated the importance of neural network algorithms in the study of carbonate slides. Weixing et al. carried out image enhancement of rock fractures based on the fractional differential[18]. They showed that the gray scale algorithm is very important in the analysis of carbonate pore

what is the context?

reply: The red line text have been deleted. The yellow text is changed to:

This paper studies the acquisition of rock images through rock slides, and the use of neural network algorithms to process the rock images to achieve rock image classification recognition based on probabilistic neural networks.

The above red text will be modified by the polisher.

cite:

The text should provide context for the information referenced here. How the information is related to the approach adopted?

Pak et al. studied the 3D imaging of a previously unidentified pore-scale process during multiphase flow in porous media[20]. They showed that it is feasible to use mathematical methods to simulate the pore distribution of carbonate rocks in 3D. Seyyedi et al. studied the pore structure changes that occur during the injection of CO₂ into carbonate reservoirs[2]. They showed that the pores in carbonate rocks may indeed be affected by microbial chemical reactions. Ghiasi-Freze et al. studied the automated Dunham classification of carbonate rock using image processing[22]. They showed that it is feasible to apply binary image analysis in the study of carbonate pores. Goral et al. studied the nanofabrication of synthetic nanoporous geomaterials[23]. They showed that the Euler number plays an important role in the study of rock pores. Kofz et al. studied the fabrication of arbitrary 3D suspended hollow microstructures in transparent fused silica glass[24]. They showed that the Euler number has important uses in the computer simulation of 3D structures. Ishayev et al. studied the use of resin-based 3D printing to build geometrically accurate models for porous sedimentary rocks[25]. They showed that Euler number classification is very important in the computer 3D reconstruction of carbonate pores. Gohrethan et al. conducted improving preservation state assessment of carbonate microfossils in paleontological research using label-free stimulated Raman imaging[26]. They showed that image processing technology has a wide range of applications in carbonate rock research. Goral et al. studied the nanofabrication of synthetic nanoporous geomaterials from nanoscale-resolution 3D imaging to nano-3D-printed digital rocks[27]. They showed that 3D demonstration of the nano-scale carbonate structure can be achieved with the help of imaging technology. Berg et al. studied industrial applications of digital rock technology[2]. They showed that digital imaging technology can be applied to the field of carbonate pore analysis. Ali et al. conducted geophysical imaging of an ophiolite structure in the United Arab Emirates[29]. They showed that geosience images can be used to study the internal structure of rocks. Lanari et al. studied XMapTools, which is a Matlab®-based graphic user interface for microprobe quantified image processing[30]. They demonstrated the important role that MATLAB in image processing. Yarmohammadi et al. conducted reservoir microfacies analysis

reply: On the basis of this review opinion and the opinions of other reviewers, this part has made relatively large changes.

cite:

exploiting microscopic image processing and classification algorithms applied to carbonate and sandstone reservoirs[31]. They showed that it is necessary to apply polarized light microscopy in the study of carbonate rocks. Kurz studied hyperspectral image analysis of different carbonate lithologies[32]. He showed that the application of image grayscale technology to carbonate rock research is feasible. Fusi et al. studied mercury porosimetry as a tool for improving the quality of micro-CT images of low porosity carbonate rocks[33]. They showed that it is feasible to use image technology from multiple sources in carbonate research. Harris studied a case of lanthanum carbonate ingestion thought to be phlebosclerotic colitis using CT imaging and abdominal radiograph[34]. He showed that the image finite automata algorithm is very important to carbonate rock research. Maheshwari compared carbonate HCl acidizing experiments with 3D simulations[35]. He showed that the traditional methods of studying carbonate rocks can be combined with computer technology. Munoz et al. studied pre-peak and post-peak rock strain characteristics during uniaxial compression using 3D digital image correlation[36]. They showed that image technology can be used to study the uniaxial compressive strength of rocks. Sharafisafa conducted an experimental investigation of the dynamic fracture patterns of 3D printed rock-like materials under impact using digital image correlation[37]. They showed that it must be compared with traditional experimental methods when using imaging technology for rock research. Kim conducted stress estimations via deep rock core diametrical deformation and joint roughness assessment using X-ray CT imaging[38]. She showed that the Euler number can be classified using image processing technology in rock pore research. Schepp et al. studied digital rock physics and laboratory considerations for a high-porosity volcanic rock[39]. Their study provides a new idea for improving the traditional method of carbonate porosity research. Saenger et al. analyzed high-resolution X-ray computed tomography images of the Bentheim sandstone under elevated confining pressures[40]. Their results provide a new image processing

Why the information is referenced here?
why it is important for the context of the research?

reply: On the basis of this review opinion and the opinions of other reviewers, this part has made relatively large changes.

cite:

STUDY OF CARBONATE ROCK PORE PERMEABILITY.

The Jinping area is surrounded by the Yalong River and contains extensive carbonate formations. The Baidiao area is a typical area containing carbonate rocks in the Jinping area. The local carbonate stratum has a deep burial depth and a large number of pores. A uniaxial pressure test conducted on the local carbonate found that the porosities of the rock samples from many of the formations increased and the uniaxial pressure resistance decreased. The decrease in the uniaxial compressive strength of the local carbonate rocks may be caused by microorganisms in the karst water. Whether this conjecture is correct or not can be determined via 16S rDNA testing. The uniaxial compressive strength of the local rocks is closely related to their porosity. Therefore, it is necessary to develop a low-cost method for researching carbonate pores.

You should provide more information on the geological setting. The text is confusing and need to be completely rewritten.
The information about the uniaxial test and the influence of microorganisms on the variation of the rock mechanics is confusing. This sections contains information about the results? And conclusion?

reply: The original text of this paragraph is very long and has been deleted a lot. When translating Chinese into English, the translation is based on the deleted text. Some texts may be puzzling to read. This paragraph has been rewritten in accordance with this opinion:

The Jinping area is surrounded by the Yalong River and contains extensive carbonate formations. As a result of the construction of hydropower projects, extensive geological surveys have been carried out in the Jinping area. In order to ensure the accuracy of the geological survey, extensive drilling and sampling have been carried out in the Jinping area. Jingfengqiao and Baidiao areas are also typical carbonate

distribution areas in Jinping area. Wells are distributed in both locations, so deep carbonate rock samples in both locations are easy to obtain. Due to the needs of engineering construction, the rock samples collected in the two places need to use the traditional carbonate research method (TCRM) for karst research. The basic parameters of carbonate rocks obtained by karst research are good verification standards. The Baidiao area is close to the Yalong River, and it is easier to collect water samples from the Yalong River, and it is convenient to use 16S rDNA technology for karst water research. Therefore, the Jingfengqiao-Baidiao area is an ideal area for the study of rock porosity using carbonate microscopic images.\\

cite:

2. Experimental materials and research methods

(a) Experimental materials

In this study, 11 rock samples and 2 water samples (BD1 and BD2) were collected in the Hebian area, Baidiao, and were transported to Chongqing within 27 hours. The 11 rock samples were collected from different formations. Three of the rock samples were broken into pieces during processing. Eight of the rock samples were successfully processed into the required shape. Five of the rock samples were successfully processed into glass slides.

Samples were collected from well cores, during drilling operations?

What are the formations from which the samples came? What is the description of these carbonate rocks?

What is the "required shape" of the samples? Glass slides refers to thin sections? How the thin sections were prepared?

reply: This section was checked before submission. The drilling information of Baidiao cannot be made public. Therefore, this article uses samples from the Jingfengqiao area whose burial depth can be published instead of the rock samples from the Baidiao area. The following is the rewritten content:

In this study, Three rock samples at the depths of 69.80-71.01m were collected from borehole(No: Qx404) in Jingfengqiao area, 2 water samples (BD1 and BD2) were collected in the Hebian area, Baidiao, and water samples of Baidiao were transported to Chongqing within 27 hours. Rock samples of Jingfengqiao were transported to Chongqing within 480 hours. The rock samples were collected from T_{2z} formations. The thickness of the T_{2z} formation in Jingfengqiao area is about 150m-700m. The rock sample numbered JP12 is griotte, with a $CaCO_3$ content of about 95% . Therefore JP12 is a very pure carbonate rock. This paper uses JP12 rock samples to process two rock slides. The porosity of the JP12 rock samples obtained by the TCRM method is 0.19%.\\

The following content is unauthorized content and is only for reviewers' reference. The image pixels have been reduced and part of the content has been obscured, and cannot be regarded as submitted content:

research area and sample point:

sample of jp12:

Traditional Carbonate Research Method (TCRM) Specimen: (including other regions):

Already used for other purposes, for the reviewer's reference only

Carbonate Rock glass Slide:

Not authorized, only for reviewer's reference

cite:

(i) Traditional method of calculating rock porosity

The traditional method of calculating rock porosity is easily understood. The rock specimen is placed in water for 72 hours to soak the pores of the rock with water. After soaking the rock specimen, it is taken out of the water and placed in a cool place for 1-2 hours to air-dry the surface moisture. The rock specimen is weighed and placed in an oven to dry for 24 hours. After drying, the rock specimen is weighed again. The two weights are subtracted and divided by the density of water to obtain the volume of the rock pores. Because the rock samples used in this study have other uses, all of the rock samples were standard cylinders with a diameter of 50 mm and a height of 3 mm. Thus, the volume of the rock sample was easily calculated. In this study, the porosity of the rock sample was obtained by dividing the volume of the pores by the volume of the rock sample.

This is not a standard method used by the industry to quantify porosity in rock samples. It is used to verify density, and by the concept can be used to estimate the volume of empty space of rock samples. However this is inaccurate for porosity measurements in rocks with macroporosity and complex capillarity.

reply: In the traditional carbonate research methods (TCRM), there are many ways to study rock porosity. But in the Jinping area, this method is used the most. Because ovens are not everywhere, some engineers and workers use gasoline immersion method to calculate porosity(The accuracy of porosity measured with gasoline is not as high as with water.). The funding for this research was limited, so other more precise and expensive methods were not used. Different carbonate rock formations in Jinping area have multiple rock porosity test results in different periods. The methods used for these tests are different. These tests are also done by researchers from different departments. The porosity value calculated by the carbonate immersion method used in this paper is very close to the historical observation value in same place. If the result of this method is very different from the historical porosity, this technology cannot be used in Jinping for a long time.The first author originally wanted to put the TCRM method research results of different strata in history into the database. In this way, the porosity of the rock obtained by the image analysis method can automatically find the nearest historical porosity of the same formation in database according to the coordinates, without organizing the experiment of the traditional carbonate research method.

cite:

(b) Using finite automata to construct a 3D data model of carbonate rock

It is no clear if the work used thin section images, or CT images.

The Euler number can be used to classify the pixel RGB values of the polarized image of the rock. In this study we used the following formula to obtain the Euler number using the RGB

reply: I edited this sentence to:

The Euler number can be used to classify the pixel RGB values of rock glass slide images.

c ite:

$$[E_n] = \begin{cases} sechR = \sum_{n=0}^{\infty} \frac{(-1)^n}{n!(n+G+B+E_n)} \left(\frac{n}{2}\right)^{2n+G+B}, & E_n(R_n) \geq E_n(G_n), \\ sechG = \sum_{n=0}^{\infty} \frac{(-1)^n}{n!(n+R+B+E_n)} \left(\frac{n}{2}\right)^{2n+R+B}, & E_n(G_n) \geq E_n(R_n), \\ sechB = \sum_{n=0}^{\infty} \frac{(-1)^n}{n!(n+G+R+E_n)} \left(\frac{n}{2}\right)^{2n+G+R}, & E_n(B_n) \geq E_n(R_n), \\ Gray = R \times 0.299 + G \times 0.587 + B \times 0.114 \end{cases} \quad (2.1)$$

According to the Euler number results, the rock polarized image was divided into nine levels according to the RGB value. In the same polarized image of the rock, the lower the Euler number classification, the lower the porosity of the area, and the higher the Euler number classification, the higher the porosity of the area. The classified image was used as the texture of the 3D cube and was pasted it onto each surface of the cube, and then, 3D models of the carbonate rock could be obtained.

If the samples are cylindrical, why the work tried to build a cubic model? The images were obtained from thin sections, and after that processed to reproduce a cube? What is the rationale applied here? This stage of the work/methodology is completely missing.

reply: In order for readers to better understand the author's purpose, the following instructions are added to the paper:

In the construction of the Jinping area, the water permeability of the rock in front of the construction is very important for construction safety. The traditional solution is to

make a small artificial cave in front of the construction route. Since not all specimens during construction can be studied by the traditional carbonate research method (TCRM), sometimes engineers must immediately determine the porosity of the rock ahead of the engineering route. The rock specimens collected in the small artificial cave had to be judged by engineers with naked eyes and experience. The accuracy of this method is not high, and it is difficult to improve the accuracy even if more engineers are recruited. So as shown in Figure 1, this paper hopes to use the classified image as the texture of the 3D cube, the classified image is obtained by classifying the RGB values of the rock image according to Euler number, and display all the visible surfaces of the cube with the classified image, so as to obtain the 3D model of the carbonate rock. This paper hopes that such a cube can help engineers better judge rock porosity.

The following content has not been approved for publication in this article, and is only for reviewers' reference:

With rock samples like the following, engineers must use their own experience to judge the porosity of the rock in about 2 days:

If the judgment is wrong, the consequences are serious.

cite:

Figure 1. Using the image processing methods to obtain 3D models of the carbonate rocks

The graph is confusing. Please inform the scale of the images, the pixel scale, the are imaged...?

This figure is not mentioned in the text above(?)

The graph does not make clear how the treated images (R, G and B channels of each image) are used to populate the cube?

reply: All microscopic images of rocks in the Jinping area had their scales removed before publication. This picture is quoted in the newly revised text. The newly revised text adds a description:

So as shown in Figure 1, this paper hopes to use the classified image as the texture of the 3D cube, the classified image is obtained by classifying the RGB values of the rock image according to Euler number, and display all the visible surfaces of the cube with the classified image, so as to obtain the 3D model of the carbonate rock. This paper hopes that such a cube can help engineers better judge rock porosity.

cite:

reply: The author is not ignoring the opinions of the reviewers, but thin sections are different from glass slides in traditional carbonate rock research. It seems that the translation into glass slides here is more accurate? The translation software is recommended to be translated into "polarizing light microscopy". The author decided to accept the reviewer's suggestion here and change it to "petrographic microscope". "The JPG images were converted to grayscale images through programming. "Explain as follows:

JPG images follow the standard gray-scale formula and realize the gray-scale processing of the image in the way of c# programming. All the programming codes used in this paper have been uploaded.

The above content is not the final text, it is finalized after polishing.

cite:

Please use numbering of the images to provide a description of each processing stage. What is the scale of the images (pixel scale).

Figure 2. Using image processing methods to obtain the porosity of carbonate rocks

reply: number of image had been added:

Figure 2. Using image processing methods to obtain the porosity of carbonate rocks

A:Original image;B:Photoshop preprocessing of the original image;
C:B picture after grayscale processing;D:C picture after finite automata image processing;
E:D picture after processed with imageJ2X.

This article has not obtained permission to use scale bars in all images, although it is easy to do so.

cite:

You need to explain the procedures for the threshold definition

In order to ensure the correctness of the finite automata, multiple image processing threshold mappings were used when establishing the finite automata. The image processing threshold

reply:I explain the procedures for the threshold definition by following text:

The characteristic of finite automata is the finiteness of mapping. This characteristic has obvious advantages in image threshold processing. The distribution range of image pixel RGB value is [0,255], a total of 256 mappings. If this paper uses Euler's number to filter RGB values in finite automata, the number of RGB values conforming to Euler's filter mapping will be greatly reduced. In this paper, all RGB values that meet Euler's number filter mapping are used as thresholds for image threshold processing. The porosity obtained by each threshold is compared with the porosity obtained by the traditional carbonate research method (TCRM) to determine the accuracy of the threshold.

The above content is not the final text, it is finalized after polishing.

cite

$$[DFA \ M] = \left\{ \begin{array}{ll} k \in [gray(i,j), R(i,j), G(i,j), B(i,j), En(i,j)], & i \in [0, pic1.width], j \in [0, pic1.height] \\ \sum \in [T_1(i,j), T_2(i,j), T_3(i,j), T_4(i,j), T_5(i,j)], & f \in [q_1(i,j), q_2(i,j)] = (0, 0, 0) \in \sum \\ T_1(i,j) = gray(i,j) > s, s \in [0, 255] \in k, & T_1(i,j) \rightarrow A(i,j), A(i,j) = (255, 255, 255) \\ T_2(i,j) = R(i,j) > s, s \in [0, 255] \in k, & T_2(i,j) \rightarrow A(i,j), A(i,j) = (255, 255, 255) \\ T_3(i,j) = G(i,j) > s, s \in [0, 255] \in k, & T_3(i,j) \rightarrow A(i,j), A(i,j) = (255, 255, 255) \\ T_4(i,j) = B(i,j) > s, s \in [0, 255] \in k, & T_4(i,j) \rightarrow A(i,j), A(i,j) = (255, 255, 255) \\ T_5(i,j) = En(i,j) > s, s \in [0, 532] \in k, & En(i,j) \rightarrow A(i,j), A(i,j) = (255, 255, 255) \\ z \in [k, f] \rightarrow \sum \cup [q_1(i,j), q_2(i,j)], & z \rightarrow [(0, 0, 0), (255, 255, 255)] \\ q_1(i,j) = z + 6 + (1 - \frac{k}{255}) \times 0.5, & f = 0 \rightarrow f = 1, A(i,j) = (255, 255, 255) \end{array} \right. \quad (2.2)$$

The text need to explain all the terms used in the Equations!

reply:the text had been added:

rsos.royalsocietypublishing.org R. Soc. open i

$$[DFA \quad M] = \left\{ \begin{array}{ll} k \in [Gray_{(i,j)}, R_{(i,j)}, G_{(i,j)}, B_{(i,j)}, E_n_{(i,j)}], & i \in [0, pic1.width], j \in [0, pic1.height] \\ \Sigma \in [T_1_{(i,j)}, T_2_{(i,j)}, T_3_{(i,j)}, T_4_{(i,j)}, T_5_{(i,j)}], & f \in [q_1_{(i,j)}, q_2_{(i,j)}] = (0, 0, 0) \in \\ T_1_{(i,j)} = gray_{(i,j)} > s, s \in [0, 255] \in k, & T_1_{(i,j)} \rightarrow A_{(i,j)}, A_{(i,j)} = (255, 255, 255) \\ T_2_{(i,j)} = R_{(i,j)} > s, s \in [0, 255] \in k, & T_2_{(i,j)} \rightarrow A_{(i,j)}, A_{(i,j)} = (255, 255, 25) \\ T_3_{(i,j)} = G_{(i,j)} > s, s \in [0, 255] \in k, & T_3_{(i,j)} \rightarrow A_{(i,j)}, A_{(i,j)} = (255, 255, 25) \\ T_4_{(i,j)} = B_{(i,j)} > s, s \in [0, 255] \in k, & T_4_{(i,j)} \rightarrow A_{(i,j)}, A_{(i,j)} = (255, 255, 25) \\ T_5_{(i,j)} = E_n_{(i,j)} > s, s \in [0, 532] \in k, & E_n_{(i,j)} \rightarrow A_{(i,j)}, A_{(i,j)} = (255, 255, 25) \\ z \in [k, f] \rightarrow \Sigma \cup [q_1_{(i,j)}, q_2_{(i,j)}], & z \rightarrow [(0, 0, 0), (255, 255, 255)] \\ q_1_{(i,j)} = z + 6 + (1 - \frac{k}{255}) \times 0.5, & f = 0 \rightarrow f = 1, A_{(i,j)} = (255, 255, 25) \end{array} \right. \quad (2.2)$$

k is a finite state set ; $Gray_{(i,j)}, R_{(i,j)}, G_{(i,j)}, B_{(i,j)}, E_n_{(i,j)}$ are finite state of k ; Σ is map list; $T_1_{(i,j)}, T_2_{(i,j)}, T_3_{(i,j)}, T_4_{(i,j)}, T_5_{(i,j)}$ are maps of Σ ; i, j is the row and column number of the

pixel; f is the result set of Σ ; $q_{(i,j)}$ is the mapping result of Σ ; $A_{(i,j)}$ is new value of color value; z is the result state set.

cite:

(d) Analysis of the causes of rock pores

The results of the finite automaton image analysis method and the traditional research methods both showed that the samples of the carbonate formations in the Baidiao area have high porosities. The reason for the high porosities of the samples of the carbonate strata in the Baidiao area is worthy of attention. The Baidiao area is located in the Yalong River Basin. The Yalong River is the largest source of karst water in the Baidiao area. **If the Yalong River water contains microorganisms such as nitrifying bacteria, sulfobacteria, and *Thiobacillus denitrificans*, the microorganisms in the Yalong River water may enter the karst water through the pores of the porous carbonate rock formations.** The microorganisms in the Yalong River may produce nitrification or sulfidation in order to maintain their own survival, thereby changing the amount of hydrogen ions in the karst water, affecting the karst process of the carbonate rock formations, and expanding the pores. **To determine whether the above hypothesis is correct, we collected water from the Yalong River in the Baidiao area for 16S rDNA analysis. If microorganisms such as nitrifying bacteria, sulfobacteria, and *Thiobacillus denitrificans* are detected in the Yalong River in the Baidiao area, the above hypothesis is convincing.**

What was the depth of sampling of the studied rocks? The assumption that superficial waters enters the studied formations need to be better addressed. The formulation of the hypothesis about the relationship between the presence of bacteria in the superficial waters and the formation of pores in the buried carbonate rocks is very confusing, and it has no relation with the other part of the research. Furthermore, finding bacteria which could be associated with carbonate dissolution in superficial waters does not means that the porosity was created only by their activity in formation waters.

3. Results

reply:the depth of rock sample had been told in " Experimental materials",The reply to " The assumption that superficial waters enters the studied formations need to be better addressed.":

There is a wide distribution of pores in the carbonate formations in the Jinping area. The local engineering department used isotope tracing technology to test pores in the carbonate formations in the Jinping area. The research in this paper is based on the conclusion of the isotope tracer report.

流态类型	埋藏深度(米)	流态成因模式	流态特态
第三亚态			
第二亚态		For reviewers' reference only	
第一亚态	00 余		

The reply to "The formulation of the hypothesis about the relationship between the

presence of bacteria in the superficial waters and the formation of pores in the buried carbonate rocks is very confusing, and it has no relation with the other part of the research.":

The key formula for the chemical reaction between carbonate rocks and microorganisms in the Jinping area has been explained above. The first author wants to explain the reason for the pore expansion of carbonate rocks in Jinping area. The expansion of the pores of carbonate rocks by microorganisms is also very important for this study.

The reply to "Furthermore, finding bacteria which could be associated with carbonate dissolution in superficial waters does not mean that the porosity was created only by their activity in formation waters.":

This article mainly focuses on the application of computer rock imaging technology in rock pore research, so 16S RDNA technology is a verification method to analyze the causes of pore formation. The detailed influence of microorganisms on karstification in Jinping area is the research content of another paper.

cite:

were used as the image processing thresholds to perform the black-and-white binarization of the rock images of T_{2y}^3 and to calculate the porosity. The porosity results obtained using the finite automata method were compared with the results of the traditional method to obtain the image processing threshold. By using the 21 mapping values of the gray and RGB values corresponding to the Euler numbers as the image processing threshold, the curves shown in Figure 3 were obtained.

Explain the terms used in the equations and discretization:

reply: The reply to "Explain the terms used in the equations and discretization":

The terms in the equation have been explained in the previous formula. This paragraph has also been modified.

cite:

The explanation is very confusing. The text concluded that the thresholds are not accurate for the porosity determination. Figure 3 shows the porosity curves obtained using the gray value and RGB value separately. We found that although the porosity distribution ranges of the porosity curves are different, the trends of the four porosity curves are consistent. The porosity distribution ranges of the four porosity curves are very large, indicating that the image processing threshold of the finite automata must use porosity research results obtained using other methods as a reference. At this time, the porosity research results obtained via traditional methods are a good reference. By putting the four porosity curves together, we obtained Figure 4.

reply: According to the opinions of other reviewers, the old FIG3 was deleted. This paragraph has also been greatly rewritten.

cite:

Figure 4. Image processing threshold curve comparison

Be clear - the analysis of an image obtained from a thin section can not establish the porosity of a rock. The estimation should be based on tens or hundreds of images from a large number of thin sections. It depends on the nature of the rock, its texture, nature of porosity and the thickness of the interval of interest.

Specify which color is linked to each parameter.

reply: I think I need to explain: As discussed in the previous research method, In the traditional carbonate research method (TCRM), calculating rock porosity is a three-dimensional problem. However, when the rock glass slide is calculated by the image threshold analysis method, it is treated as a 2D plane problem. WHY? Because the thickness of the rock glass slide is very thin, it can be approximated that the height of all pores in the glass slide is the same (Approaching 0 but not 0). Therefore, it can be approximated that the ratio of the total number of pixels representing pores in the glass slide to the total number of pixels is the porosity of the glass slide. Therefore, if an algorithm can be found to correctly identify the pixel of the rock pores, the porosity can be obtained through image processing.

The old figure 4 (new figure 3) has been modified based on this recommendation and the recommendations of other reviewers.

cite:

Which trend?

We found that the trends of the four porosity curves are exactly the same.

reply: Add the following explanation to this sentence:

When all 4 curves are placed in the same coordinate system, we can find that when the value on the horizontal axis is close to the middle value of the [0,255], the value on the vertical axis is relatively large; when the value on the horizontal axis is close to the two ends of the [0,255], the value on the vertical axis is relatively small.

cite:

(b) Results of the finite automata image analysis method

This explanation includes information which should be in the methodology section

In this study, each pixel of the carbonate polarized image was converted into a gray value or RGB value. The Euler number formula used in this study has 9 sets of 20 solutions to the gray value and RGB value of the carbonate polarized image. If we put these 20 solutions into [0,255], there will be 20 image processing thresholds for the gray value or RGB value. The 20 image processing thresholds were all processed using the finite automata. For each polarized image of the carbonate rocks, 20 black and white binary images were obtained using the image processing thresholds of the gray value or RGB value. The porosity was obtained by dividing the number of black pixels by the total number of pixels in the black and white binary images. The porosity of the rock in the polarized image must be greater than 0 and less than 100%. If the porosity is less than 0 or greater than 100%, there is an error in the finite automaton. According to the above method, it was found that when the gray value was 71, the porosity was the closest to the porosity obtained using the traditional method. The black and white binary images obtained using each image processing threshold were listed, and the images shown in Figure 5 were obtained.

Methodology...

reply: Okay, this part of the content has been cut into the research method.

cite: What is the scale of the images (photo micrographies)? What is the size of the area sampled and processed here.

reply: The original scale is deleted before submission.

cite: Same observations as for the Figure 5.

reply: Figures 4, 5, 6, and 7 are the results of applying 20 thresholds on RGB and gray values, which are different.

cite:

(c) 3D data model of the carbonate rocks from Baidiao

In this study, there were 9 sets of solutions for the polarized images of the carbonate rocks from the Baidiao area at [0,255]. In addition, for the initial state when the Euler's number is 0, there are 10 Euler numbers in the Euler number formula for the Baidiao area between [0,255]. All of the Euler numbers, gray values, and RGB values were put into the same coordinate system, and the curve shown in Figure 10 was obtained.

?????

Figure 10. Euler number and image processing threshold Cross correlation?

reply: The editor does not understand it, and the reader will not understand it two. The author repeats this paragraph with Jing Fengqiao's data. The 3D model is as follows:

Figure 8. 3D data model of the carbonate rocks from Jingfengqiao

If this paragraph is too brief, it will not be easy to understand, and if it is expanded, it will be too long, so I delete this paragraph.

cite:

Methodology must be placed in the correct section.

(d) Results of the 16S rDNA analysis

Microbial community genomic DNA was extracted from water samples from the Baidiao area (bd1 and bd2) using TransGen AP221-02 according to the manufacturer's instructions. The DNA extracted was checked using 1% agarose gel, and the DNA's concentration and purity were checked using NanoDrop 2000. The hypervariable region of the bacterial 16S rRNA gene was amplified using an ABI GeneAmp®9700 PCR thermocycler. The purified amplicons were pooled in equimolar and paired-end sequenced using the an Illumina MiSeq PE300 platform/NovaSeq PE250 platform (Illumina, San Diego, USA) according to the standard protocols provided by Majorbio Bio-Pharm Technology Co. Ltd. (Shanghai, China). Based on the results obtained using the above research methods, the water samples collected from the Baidiao area do not contain nitrifying bacteria.

reply: Okay, most of this paragraph has been adjusted to research methods.

cite:

4. Discussion and analysis

(a) Comparison with the research results obtained using traditional methods

The carbonate porosity obtained using the image threshold processing method and the Euler's number method and the carbonate porosity obtained using the traditional method were put in the same coordinate system and were compared using a histogram (Figure 13). As can be seen from Figure 13, the carbonate porosities of T_{29}^5 and T_{29}^1 obtained using the three methods are relatively close. The differences in the carbonate porosities obtained using the three methods of T_{29}^{5-1-2} are also acceptable. Based on the five samples, the accuracy rate exceeds 40%. The accuracy rate is higher than that of direct judgment using the human eye. This is higher than the accuracy rate of human judgment in the engineering department in Baidiao (about 20%).

Rock porosity is not defined through visual description of thin sections. It provides only approximation, which is not used as a precise information in the characterization of reservoir rocks.

reply: In order to better highlight the comparison between the results of the image analysis method and the results of the traditional carbonate research method (TCRM), the new figure retains two methods "Image analysis method and traditional carbonate research method (TCRM)". In the new image, a rock pore image with a gray threshold value of 27 has been added.

Figure 9: Carbonate pore map based on image threshold analysis

Figure 10: Comparison of carbonate pore value obtained by image threshold analysis method and TCRM(A:carbonate pore value obtained by image threshold analysis method;B:carbonate pore value obtained by TCRM)

cite:

processing algorithms applied to the study of carbonate porosity. The number of polarized rock images used in this study is relatively small. If the number of polarized rock images used in the study were increased, there is still room for improvement in the image processing algorithm.

reply: This sentence is rewritten as follows:

The polarized rock images used in this study are not much.

cite: You should discuss the results in contrast with other works which performed similar analyses. Conclusions should be kept for the adequate section.

reply: The following description is added to this paragraph:

The $T^{\{2z\}}$ formation in Jingfengqiao area has conducted TCRM porosity studies many times. If the TCRM porosity is averaged, the average value is closer to the results of this paper. If the maximum and minimum porosity values of the TCRM of the $T^{\{2z\}}$ formation in Jingfengqiao area are used to establish the interval, the research results of the finite automata in this paper are to the right of the interval.

cite: Use the caption and the text above to explain the information in this graph.

reply: The above has made relatively large changes, including the requirements of this opinion.

cite:

(b) Using Euler numbers to build a carbonate pore model

In this study, the Euler number of the polarized image of the carbonate rock was put in a three-dimensional coordinate system to determine the intersection points, and these intersection points were represented by cubes to establish a three-dimensional pore model of the carbonate rock. In order to control the amount of calculations required, a 3×8 matrix was used to organize the Euler numbers, and the number of intersection points was kept below eight. These cubes represent the pores in the carbonate rocks. Since the results of the image threshold method

Methodology

reply: The yellow text was revised and moved to the research method.

The solution of formula 2.1 has a set of distributions in $[-50,50]$, the Euler number of the polarized image of the carbonate rock in $[-50,50]$ are arranged in ascending order, descending order, and random to set up a 3×8 matrix. Each row of the matrix is recorded as the spatial coordinate value to construct the nodes of the 3D pore distribution model in carbonate rocks. In order to control the amount of calculations required, the number of nodes was kept below eight. The nodes in these cubes represent the distribution positions of pores in the rock.

The following explanation has been added to this paragraph:

There are 9 sets of solutions in formula 2.1, and the set of $[-50,50]$ is the only set of solutions where all the nodes are in the cube. The distribution of rock pores will not be outside the rock, so this study only uses the solution between $[-50,50]$ to construct the cube. As can be seen from Figure 11, the cubes constructed using the Euler number nodes are greater in $T^{\{z\}}$ in the horizontal direction.

cite:

in T_{2y}^{5-1-2} in the vertical direction. Therefore, in T_{2y}^6 , T_{2y}^{5-1-1} , and T_{2y}^6 , the uniaxial compressive strength in the vertical direction is greater than in the horizontal direction. In T_{2y}^{5-1-2} , the uniaxial compressive strength is greater in the horizontal direction than in the vertical direction.

This information is completely disconnected from the discussion about the porosity, and the correlation does not make any sense in the way it is described.

reply: the reply to "This information is completely disconnected from the discussion about the porosity, and the correlation does not make any sense in the way it is described.":

In the original text, there are pores development that lead to changes in the uniaxial compressive strength of the rock, and these have been deleted. The translation is based on the remaining text, which may not be smooth. Therefore, all parts related to uniaxial compressive strength are deleted.

cite:

(c) Causes of the rock pores in the carbonate rocks in the Baidiao area

This assumption need to be addressed with evidence.

The pores in the carbonate rocks in the Baidiao area should be caused by the water from the Yalong River. The microorganisms in the Yalong River water and the microorganisms

reply: This sentence has been deleted.

cite:

process of the carbonate rocks and expand the pores in the carbonate rocks. Through the 16S rDNA analysis of the river water from the Baidiao area, it was found that there are nitrifying bacteria in the local Yalong River water.

This information is contrary to what is mentioned in the lines 56 to 58. What is the correct result?

The simplistic assumptions made are not sufficient to provide any conclusion about the relation between bacteria activity and the origin of the porosity

reply: I am sorry that when I checked the English translation, I did not find that both "nitrifying bacteria" and "denitrifying bacteria" were turned into nitrifying bacteria. Another reviewer also pointed out this error, and I corrected nitrifying bacteria and denitrifying bacteria. English translators translated both denitrifying bacteria and nitrifying bacteria into nitrifying bacteria. "Results of 16S rDNA analysis" is right, the water samples collected from the Baidiao area Do NOT contain nitrifying bacteria, There are only denitrifying bacteria and no nitrifying bacteria in the water samples in the Baidiao area.

In the Jinping area, the karst water and karst soil in Baidiao, Jingfengqiao, Qimulin, Mofanggou and other places have been studied on the karst process. In the Jinping area, the karst water and karst soil in Baidiao, Jingfengqiao, Qimulin, Mofanggou and other places have been studied on the karst process. The preceding text also explains this.

cite: You need to use references and a much better comparison between the results and similar research found in the literature. The discussion presented here is totally incipient and does not allow the reader to evaluate any contribution.

reply: Okay, this paragraph has been modified in accordance with this recommendation. In this study, engineering reports or engineering databases are actually used for comparison, which is difficult for readers to obtain, so this paragraph mainly uses publicly published paper for comparison:

Compared with the research results of Nanpu area \cite{k3}, the accuracy of this paper is acceptable. Compared with using CT to build a 3D pore model of carbonate rock \cite{k8}, the cost of this research method is lower. Compared with the research results in Taihang area \cite{k49}, the research method used in this paper is lower in cost and shorter in research period. Compared with the study of rock pores using only image processing technology \cite{k25} \cite{k27} \cite{k31}, the research results of using TCRM to verify the image finite automata used in this paper are easier to be accepted by the engineering department.